# Hydralazine induces stress resistance and extends *C. elegans* lifespan by activating the NRF2/SKN-1 signalling pathway

Esmaeil Dehghan[1], Yiqiang Zhang[2], Bahar Saremi[1], Sivaramakrishna Yadavali [1], Amirmansoor Hakimi[1], Maryam Dehghani[1], Mohammad Goodarzi[1], Xiaoqin Tu[1], Scott Robertson[3], Rueyling Lin[3], Asish Chudhuri[1] & Hamid Mirzaei [1]

Nuclear factor (erythroid-derived 2)-like 2 and its *Caenorhabditis elegans* ortholog, SKN-1, are transcription factors that have a pivotal role in the oxidative stress response, cellular homeostasis, and organismal lifespan. Similar to other defense systems, the NRF2-mediated stress response is compromised in aging and neurodegenerative diseases. Here, we report that the FDA approved drug hydralazine is a bona fide activator of the NRF2/SKN-1 signaling pathway. We demonstrate that hydralazine extends healthy lifespan (~25%) in wild type and tauopathy model *C. elegans* at least as effectively as other anti-aging compounds, such as curcumin and metformin. We show that hydralazine-mediated lifespan extension is SKN-1 dependent, with a mechanism most likely mimicking calorie restriction. Using both in vitro and in vivo models, we go on to demonstrate that hydralazine has neuroprotective properties against endogenous and exogenous stressors. Our data suggest that hydralazine may be a viable candidate for the treatment of age-related disorders.

[1] Department of Biochemistry, UT Southwestern Medical Center, Dallas, TX 75390, USA. [2] Greehey Children's Cancer Research Institute, UT Science Center at San Antonio, San Antonio, TX 78229, USA. [3] Department of Molecular Biology, UT Southwestern Medical Center, Dallas, TX 75390, USA. Correspondence and requests for materials should be addressed to H.M. (email: hamid.mirzaei@utsouthwestern.edu)

One of the main mechanisms underlying compromised physiological function in aging and age-related diseases is chronic elevation of reactive oxygen species (ROS)[1,2]. Because oxidative damage is a direct threat to cell survival, several important defense machineries (i.e., ROS scavengers, repair and refold machineries and degradation apparatus) have evolved to maintain cellular homeostasis. When these defense machineries are compromised, as observed in aging and age-related diseases (i.e., Alzheimer's (AD), Parkinson's (PD), Huntington's disease (HD), etc.) cell function is misregulated and cell death is accelerated[3,4].

Nuclear factor erythroid 2-related factor 2 (NFE2L2) or NRF2 is a master regulatory element modulating a diverse set of anti-oxidant defense machineries[5,6]. NRF2 regulates more than 200 genes encoding cytoprotective phase II detoxification and anti-oxidant enzymes, including HMOX1, NQO1, glutamate-cysteine ligase subunits (GCLC and GCLM), and glutathione-S-transferase (GST) which collectively synthesize glutathione (GSH) and assist maintaining GSH over the oxidized form GSSG[7,8].

Under normal conditions, NRF2 is sequestered in the cytosol by a KEAP1 (Keltch-like ECH associated protein 1) homodimer. The half-life of NRF2 is short (~15 min) as it is ubiquitinated and rapidly degraded by the proteasome machinery[9,10]. When cells are stressed, however, a conformational change is induced in KEAP1, mediated by three reactive cysteine residues, resulting in the release of NRF2[11]. Once released, NRF2 escapes the CUL3-mediated degradation pathway which increases its half-life to 60 min. Free NRF2 is then phosphorylated at Ser-40 by protein kinase C which triggers the translocation of pNRF2 into the nucleus[12]. pNRF2 then rapidly enters the nucleus and after reduction of its cysteines by TXN, binds to antioxidant response element (ARE) sequences in the upstream promoter regions of many antioxidant genes[13]. To develop a molecular probe for identification of carbonylated proteins in brain, we searched for a molecule that (1) reacts with protein carbonyls efficiently, (2) crosses the blood–brain barrier, (3) has a suitable structure for attachment of a purification handle, and (4) is nontoxic. We selected hydralazine because it met all the above-mentioned criteria. We discovered that this drug, FDA approved for the treatment of hypertension, has anti-aging properties. Here, we report for the first time that hydralazine activates the NRF2 signaling pathway. Using in vitro and in vivo model systems (human neuroblastoma cell line (SH-SY5Y) and *Caenorhabditis elegans*), we show that hydralazine treatment activates cyto-protective elements by triggering the translocation of NRF2 from the cytoplasm to the nucleus followed by ARE activation. We demonstrate that hydralazine extends healthy lifespan in *C. elegans* by activating SKN-1, the NRF2 ortholog in worms. Additionally, we illustrate using both in vitro and in vivo models that hydralazine protects against exogenous and endogenous stressors such as rotenone and tau aggregates. We suggest that activation of NRF2 by hydralazine provides a protective mechanism to shield neuronal cells, otherwise vulnerable in a compromised environment that elicits aging and diseases such as AD and PD.

## Results

**Hydralazine protects cells from H₂O₂ cytotoxicity.** In addition to its utility in the treatment of hypertension, hydralazine was shown to inhibit acrolein-mediated injuries in ex vivo spinal cord via acrolein aldehyde functional group chelation[14]. Considering the importance of aldehyde toxicity and the potential benefits of identifying carbonylated proteins, we first tested the reactivity of hydralazine (Hyd) with intracellular aldehydes. To generate aldehydes, we treated SH-SY5Y cells with 100 μM hydrogen peroxide ($H_2O_2$) for 24 h. Carbonyl groups were quantified using a 2,4-DNPH (dinitrophenylhydrazine) assay. Hydrazine (Hy), a compound with the same functional group as hydralazine, was used as a positive control. Control and stressed cells were both treated with 10 and 25 μM of hydralazine or hydrazine (Fig. 1a, b). Both hydrazine and hydralazine reduced protein carbonyls significantly. Surprisingly, when we assayed cell viability using an 3-[4,5-dimethylthiazol-2-yl]-2,5-diphenyltetrazolium bromide; thiazolyl blue (MTT) assay under the same experimental conditions, hydralazine protected cells from $H_2O_2$ induced cell death whereas hydrazine failed to provide protection (Fig. 1c, d). We next used 2′,7′-dichlorofluorescin diacetate (DCFDA) to quantify ROS in SH-SY5Y cells 0.5, 1, 3, 6, 12, and 24 h after hydralazine treatment to confirm that hydralazine itself does not increase endogenous ROS (Fig. 1e).

**Hydralazine activates the NRF2/SKN-1 pathway.** We next sought to identify hydralazine's mode of action by performing a global comparative proteomics screen using stable isotope labeling with amino acids in cell culture (SILAC) (Fig. 2a)[15]. SH-SY5Y cells grown in light or heavy media were treated with 0 or 10 μM of hydralazine respectively. After 24 h cells were collected and lysed. Equal amounts of heavy and light lysates were combined, digested, and analyzed by shotgun mass spectrometry which resulted in quantification of ~5400 proteins. The SILAC results were searched using the Ingenuity Pathway Analysis (IPA) tool which implicated activation of the NRF2 pathway among others (Z score = 0.156, p value = 1.97E−10) (Fig. 2b, c and Supplementary Fig. 1 and Supplementary Data 1). We pursued NRF2 because it best explained the hydralazine protective phenotype and the fact that we anticipated it to be among downregulated pathways due to hydralazine-mediated ROS decrease. The opposite effect suggested that activation of NRF2 was not oxidatively regulated.

We quantified NRF2 protein in treated cells as a first step towards validating the activation of the NRF2 pathway (Fig. 3a). SH-SY5Y cells were treated with hydralazine (0, 2.5, 5, and 10 μM) for 24 h followed by total lysate western blot analysis or immunoprecipitation (IP)–western blot analysis showing a dose-dependent increase in NRF2 signal intensity of up to 80%. Hydrazine treatment (5 μM) did not show any effect (Supplementary Fig. 2a). Antibody specificity was validated using NRF2 knockdown SH-SY5Y cells (Supplementary Fig. 2b). We also separated the NRF2 immunoprecipitates on SDS-PAGE, cut the NRF2-corresponding band, and analyzed it by mass spectrometry. Label-free quantification of NRF2 showed a similar trend as western blots. A peptide representing NRF2 was extracted from the total ion chromatogram for the manual validation (Fig. 3b). We next investigated the effect of hydralazine on the interaction between NRF2 and KEAP1, by reciprocal NRF2–KEAP1 and KEAP1–NRF2 co-immunoprecipitations. Hydralazine significantly reduced NRF2-bound KEAP1 in the anti-NRF2 antibody pull down. Similarly, KEAP1-bound NRF2 was significantly reduced in the presence of hydralazine in the anti-KEAP1 antibody pull down (Fig. 3c). We next examined NRF2 nuclear localization, the next step in NRF2 pathway activation, by performing subcellular fractionation and NRF2 partition quantification in SH-SY5Y cells. The nuclear localization of NRF2 was significantly increased in cells treated with hydralazine while the cytosolic NRF2 fraction remained unchanged (Fig. 3d). NRF2 phosphorylation on serine 40 is a regulatory modification required for NRF2 translocation to the nucleus and downstream protein activation. As shown in Fig. 3e, the phosphorylation of NRF2 on serine 40 in the nuclear compartment was increased by hydralazine treatment. No change was observed in

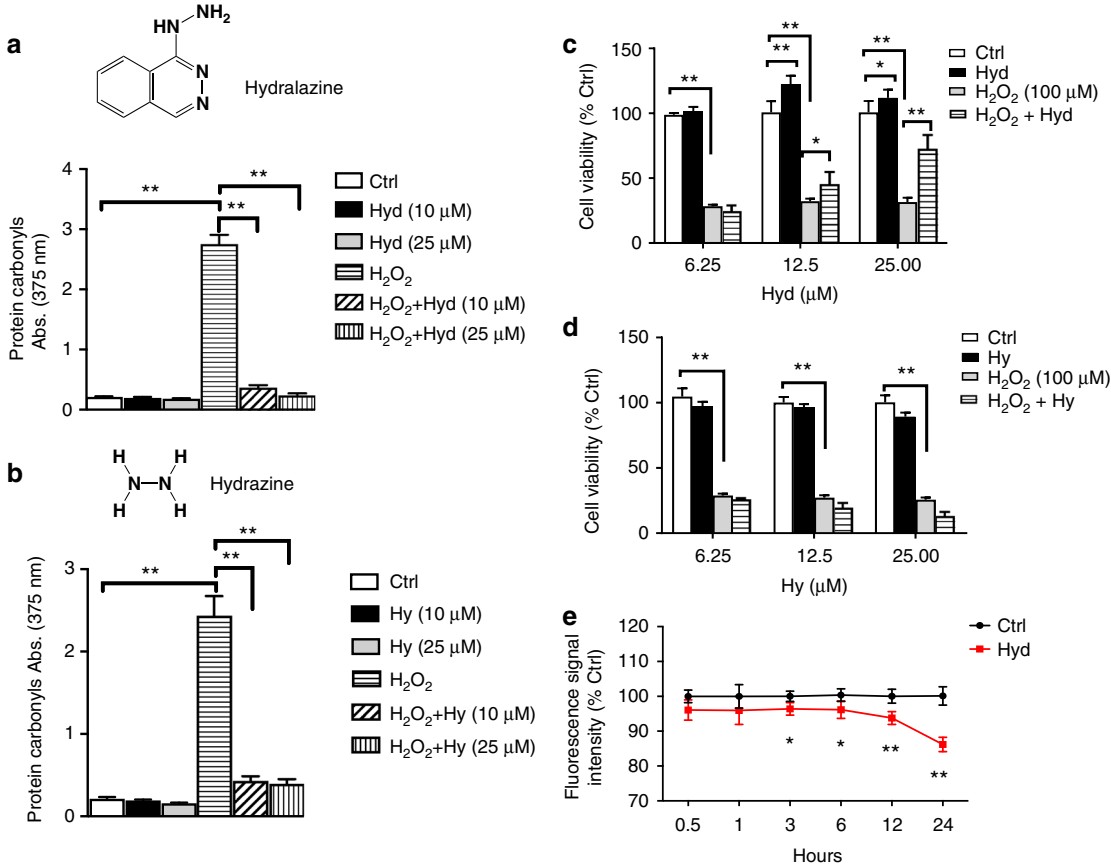

**Fig. 1** Hydralazine protects SH-SY5Y cells from oxidative stress induced cell death. **a** Protein carbonyls were measured using 2-DNPH assay in SH-SY5Y cells treated with hydralazine, $H_2O_2$, or both. Hydralazine treatment significantly reduced the carbonyl concentration raised by $H_2O_2$ treatment. **$p < 0.01$, two-tailed Student's $t$ test, $n = 6$, mean ± SD. **b** Hydrazine, a compound with the same aldehyde chelating functional group as hydralazine, reduced the concentration of carbonyls in the lysate. **$p < 0.01$, two-tailed Student's $t$ test, $n = 6$, mean ± SD. **c** The viability of cells under $H_2O_2$ induced stress was significantly improved with hydralazine treatment. *$p < 0.05$ and **$p < 0.01$, two-tailed Student's $t$ test, $n = 6$, mean ± SD. **d** Hydrazine treatment did not rescue cells from $H_2O_2$ stress, indicating that the hydralazine mode of action did not depend on carbonyl sequestration. **$p < 0.01$, two-tailed Student's $t$ test, $n = 6$, mean ± SD. **e** ROS were measured using a DCDFA assay. Hydralazine treatment of SH-SY5Y cells (10 μM) did not increase ROS 30 min after treatment and beyond, confirming that hydralazine itself does not cause oxidative stress. *$p < 0.05$ and **$p < 0.01$, two-tailed Student's $t$ test, $n = 8$, mean ± SEM

phosphorylation of cytosolic NRF2. We used lamin and actin respectively as markers for purified nuclear and cytosolic fractions.

We next quantified the relative amounts of thioredoxin (TXN) in untreated versus hydralazine-treated cells, which showed a significant increase in treated cells (Fig. 3f). TXN reduces the cysteine residues critical for binding of nuclear NRF2 to ARE[13]. To determine the functional relevance of the increases in expression, translocation and phosphorylation of NRF2, we measured NRF2 transcriptional activity using a luciferase-based ARE-controlled gene expression system. SH-SY5Y cells expressing the ARE-luciferase reporter were treated with hydralazine for 24 h prior to measurement. Hydralazine increased luciferase activity significantly compared to untreated cells ($2.0 ± 0.2$ folds) (Fig. 3g).

To further confirm NRF2 activation, we next measured the relative expression of NRF2 downstream target genes, *NQO1*, *HMOX1*, *GCLC*, *GCLM*, *GST4*, and *GSTP1* by qRT-PCR in SH-SY5Y cells treated with hydralazine for 24 h and showed they were all upregulated (Fig. 3h). We also measured the relative abundance of NRF2 downstream targets, GCLC, GCLM, HMOX1, and NQO1 by western blot analysis. All four targets showed a significant increase in expression (Fig. 3i). Uncropped scans for blots presented in Fig. 3 are shown in Supplementary Fig. 8.

The *C. elegans* NRF2 ortholog, SKN-1, shows remarkable functional conservation relative to its mammalian counterpart, making *C. elegans* an ideal model for in vivo studies of the NRF2 pathway[16,17]. SKN-1 is primarily expressed in the intestine where it regulates oxidative and xenobiotic stress responses. It is also expressed in ASI chemosensory neurons (putative hypothalamus) where it mediates the longevity benefits of dietary restriction[16,18].

Hydralazine treatment significantly increased the localization of SKN-1::GFP in the intestinal nuclei (Fig. 4a, b). Hydralazine treatment also increased the signal intensity of SKN-1::GFP in the ASI neurons (Fig. 4c). We measured the expression of glutathione S-transferase-4 (GST-4), a downstream target of SKN-1 isoform C, in a transgenic strain (CL2166) expressing GST-4 promoter driven GFP[19]. Hydralazine treatment caused a significant increase in the *gst-4p*::GFP signal 48 h after treatment (Fig. 4d) but did not do the same in worms fed *skn-1(RNAi)* or in a mutant strain (*skn-1(zu67)*) lacking a functional intestinal *skn-1* isoform c (Fig. 4e and Supplementary Fig. 3).

We measured superoxide concentration in wild-type and *skn-1(zu135)* mutant (which has loss of function mutation in all SKN-1 isoforms) worms treated with hydralazine. Superoxide concentration was decreased in wild-type worms but not in mutant worms, confirming activation of the SKN-1 antioxidant pathway with hydralazine treatment (Fig. 4f). To further support hydralazine-induced SKN-1 activation, we conducted a

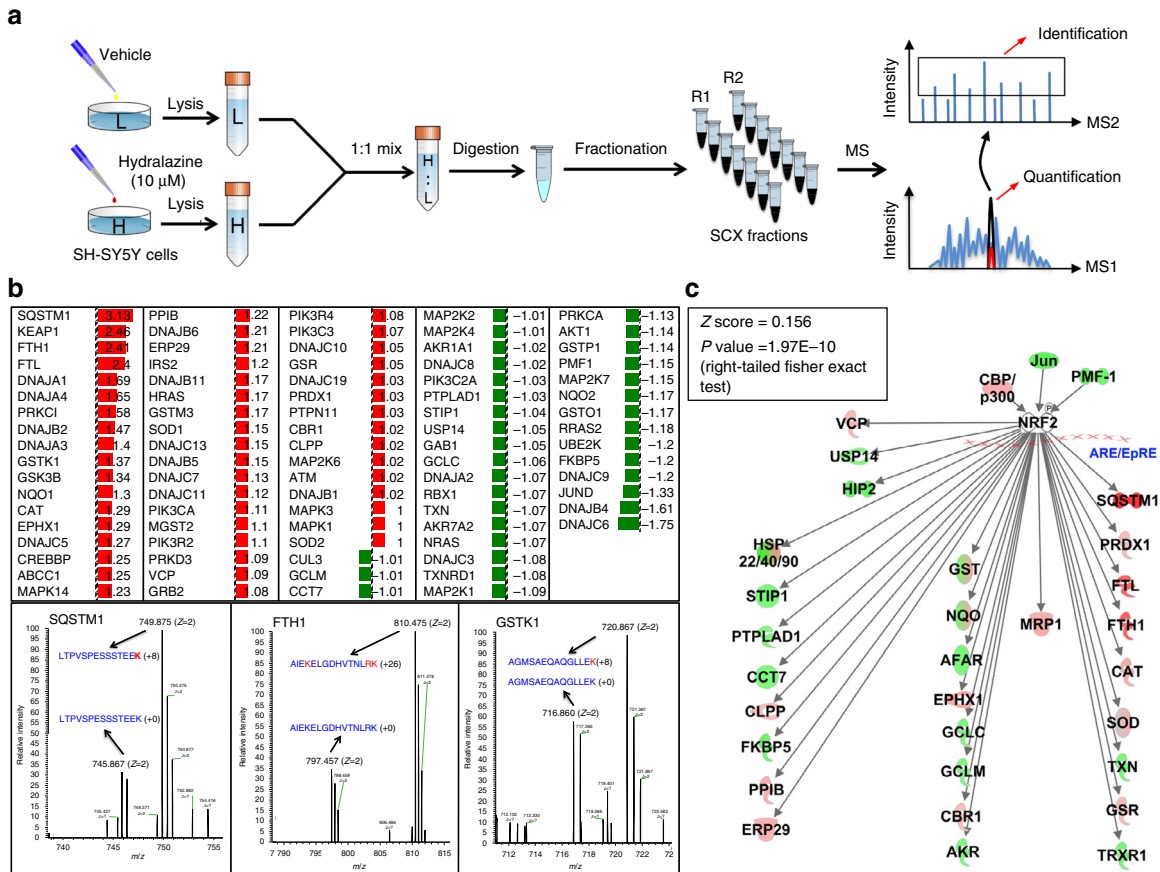

**Fig. 2** SILAC screen identifies NRF2 pathway activation with hydralazine treatment. **a** The SILAC workflow for identification of pathways activated by hydralazine (n = 2). **b** Proteins with their log2 ratios (treated/untreated) were mapped into the ingenuity pathway analysis (IPA) database where the NRF2 pathway was found activated. Raw MS data for proteins SQSTM1, FTH1, and GSTK1 are also shown as part of the manual validation of the data. **c** NRF2 pathway activation reported by IPA signified with Z score of 0.156 and p value of 1.97E−10, right-tailed Fisher exact test

global comparative proteomics analysis. Synchronized populations of wild-type *C. elegans* were treated for 3 days with 100 μM of hydralazine or vehicle. Hydralazine and vehicle-treated populations, four biological replicates each, were then lysed, digested and analyzed by shotgun mass spectrometry for label-free quantification. A total of 3113 proteins were detected across all biological replicates, of which 269 proteins were downregulated and 143 were upregulated. An IPA analysis was performed using proteins with human orthologs. The SKN-1/NRF2-stress response pathway was found to be activated with a significant score (Z score = 3.317, p-value = 0.0002, right-tailed Fisher exact test) (Fig. 4g and Supplementary Fig. 4). The SKN-1 pathway human orthologs, represented with red dots in the volcano plot and their log2 fold change (FC), are shown in Fig. 4g. These data collectively support the hypothesis generated by the quantitative proteomics screen that identified hydralazine as an activator of the NRF2/SKN-1 pathway.

**Hydralazine extends *C. elegans* healthy lifespan.** Activation of NRF2 and its orthologs are known to have prolongevity effects in various organisms[20,21]. In *C. elegans* SKN-1 expressed in ASI chemosensory neurons mediates the longevity benefits of dietary restriction by increasing metabolic activity via endocrine signaling[16,18,21]. We next set out to determine if exposure to hydralazine extends the healthy lifespan in *C. elegans*.

Synchronized populations of wild-type worms were grown on media containing hydralazine from 10 to 100 μM. A dose-

dependent lifespan extension was observed. Maximal median lifespan extension (~25%) was observed in animals treated with 100 μM hydralazine (Fig. 5a and Supplementary Fig. 5a). To rule out hydralazine-mediated alteration of microbial metabolism[22], we fed worms with bacteria pretreated with hydralazine which did not result in lifespan extension. But when worms were fed with heat-inactivated bacteria, hydralazine treatment resulted in lifespan extension (Fig. 5b). Hydralazine at the concentrations used for lifespan studies did not affect the growth rate of HB101 bacteria, further ruling out bacterial growth rate alteration as the underlying mechanism for hydralazine-mediated lifespan extension (Supplementary Fig. 5b).

We also tested the late-stage treatment efficacy of hydralazine compared to metformin (Metf) and curcumin (Curc), two well-known SKN-1 activators with anti-aging phenotypes[23,24]. Late-stage treatment with hydralazine resulted in lifespan extension similar or better than curcumin and metformin (Fig. 5c). The pro-longevity effect of hydralazine was completely lost in *skn-1* knockdown worms and *skn-1(zu135)* mutants, but not in worms fed with control vector (Fig. 5d, e). However, in mutant *skn-1 (zu67)* worms with a functional SKN-1 isoform B, partial restoration of hydralazine-mediated lifespan extension was observed (Fig. 5e). Using transgenic animals with mosaic expression of isoform *b* in the ASI neurons (*geIs9*) or isoform *c* in the intestine (*geIs10*), we confirmed the important role of neuronal SKN-1 in prolongevity effects of hydralazine. Although the presence of *skn-1* isoform *c* does not have a significant impact

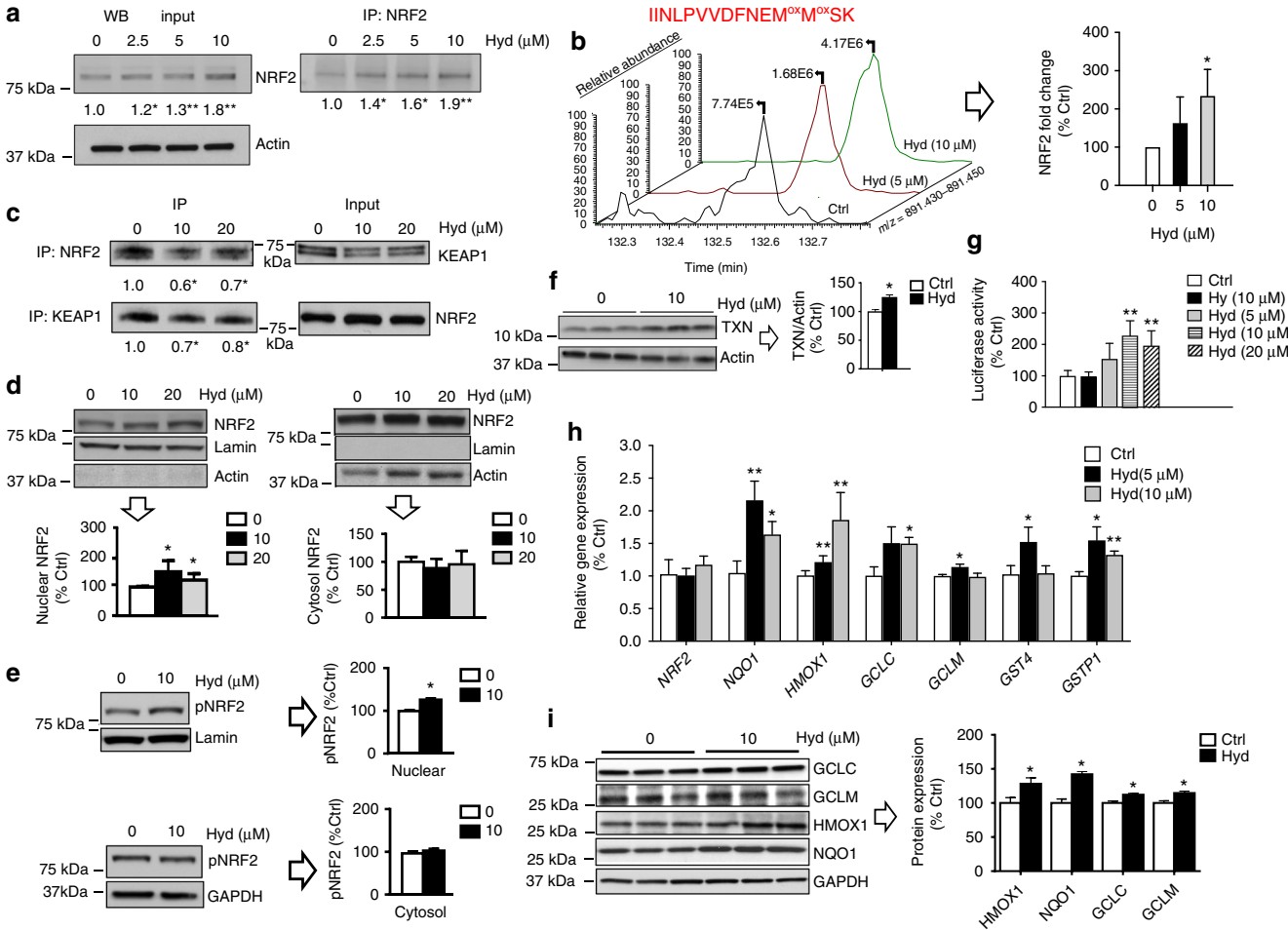

**Fig. 3** Hydralazine enhances NRF2 signaling in SH-SY5Y cells. **a** Hydralazine increased cellular NRF2 protein in a dose-dependent manner demonstrated by western blot analysis of cell lysates (input) and NRF2 immunoprecipitates (IP:NRF2). *$p < 0.05$ and **$p < 0.01$, two-tailed Student's $t$ test, $n = 3$, mean ± SD. **b** The mass spectrometry based label-free quantification of NRF2 immunoprecipitates prepared in part A. Extracted ion chromatogram of a NRF2 peptide, IINLPVVDFNEM$^{ox}$M$^{ox}$SK, as well as bar plot quantification are shown. Samples were treated with 0.05% H$_2$O$_2$ prior to mass spectrometric analysis. *$p < 0.05$, two-tailed Student's $t$ test, $n = 3$, mean ± SD. **c** Hydralazine reduced the interaction between NRF2 and KEAP1. Interactions were measured by reciprocal Co-IPs followed by western blot analysis. *$p < 0.05$, two-tailed Student's $t$ test, $n = 3$, mean ± SD. **d** NRF2 translocates to the nucleus with hydralazine treatment. Treated cells were subjected to cell fractionation and western blot analysis. *$p < 0.05$ and **$p < 0.01$, two-tailed Student's $t$ test, $n = 3$, mean ± SD. **e** Hydralazine treatment increased nuclear NRF2 phosphorylation quantified using an antibody specific to NRF2 phosphorylation at serine 40. *$p < 0.05$ and **$p < 0.01$, two-tailed Student's $t$ test, $n = 3$, mean ± SD. **f** TXN, a potent regulator of the NRF2–KEAP1 response system, is upregulated with hydralazine treatment. *$p < 0.05$, two-tailed Student's $t$ test, $n = 3$, mean ± SD. **g** ARE-driven luciferase activity was increased with hydralazine treatment, indicating an increase in the transcriptional activation of NRF2 target genes. Hydrazine did not increase luciferase activity. **$p < 0.01$, two-tailed Student's $t$ test, $n = 6$, mean ± SD. **h** Hydralazine treatment increased the expression of NRF2 downstream targets measured by qPCR using actin as internal control. For the list of primers used for qPCR, see Supplementary Table 1 online. *$p < 0.05$, and **$p < 0.01$, two-tailed Student's $t$ test, $n = 6$, mean ± SEM. **i** Hydralazine treatment increased the expression of NRF2 downstream target measured by western blot analysis. *$p < 0.05$, two-tailed Student's $t$ test, $n = 3$, mean ± SD

on lifespan extension, its availability along with isoform *b* is necessary to achieve maximum lifespan extension (Supplementary Fig. 5c).

Hydralazine also resulted in a significant improvement in the locomotor performance of young (5 days old), middle age (10 days old), and old (15 days old) wild-type animals (Fig. 5f). To further study the role of SKN-1 in hydralazine-mediated health improvement, we treated mutant *skn-1(zu135)* animals the same way as the wild type and measured their locomotion at three time points, young (4 days old), middle age (8 days old) and old (12 days old). Locomotor performance did not improve in *skn-1 (zu135)* animals with hydralazine treatment, demonstrating the role of SKN-1 in delaying age-dependent deterioration of locomotion in *C. elegans* (Fig. 5f).

We utilized a widely used dietary restriction genetic model (*eat-2*) to investigate the mechanism of hydralazine-mediated SKN-1 activation. The *eat-2* worms have a mutation in a nicotinic acetylcholine channel that reduces pharyngeal pumping and food intake, leading to an extended lifespan[25,26]. Hydralazine treatment (100 μM) did not result in lifespan extension in *eat-2 C. elegans* indicating a possible overlap between dietary restriction and hydralazine mechanism(s) of action (Fig. 5g). We measured the pharyngeal pumping rate of wild type young (day 4) *C. elegans* treated with 100 μM hydralazine to confirm that hydralazine did not inhibit food uptake by changing pharynx contraction rate (Supplementary Fig. 5d). Reduction in lipofuscin accumulation and brood size are two major phenotypes of dietary restriction[27]. To further validate our hypothesis, we measured

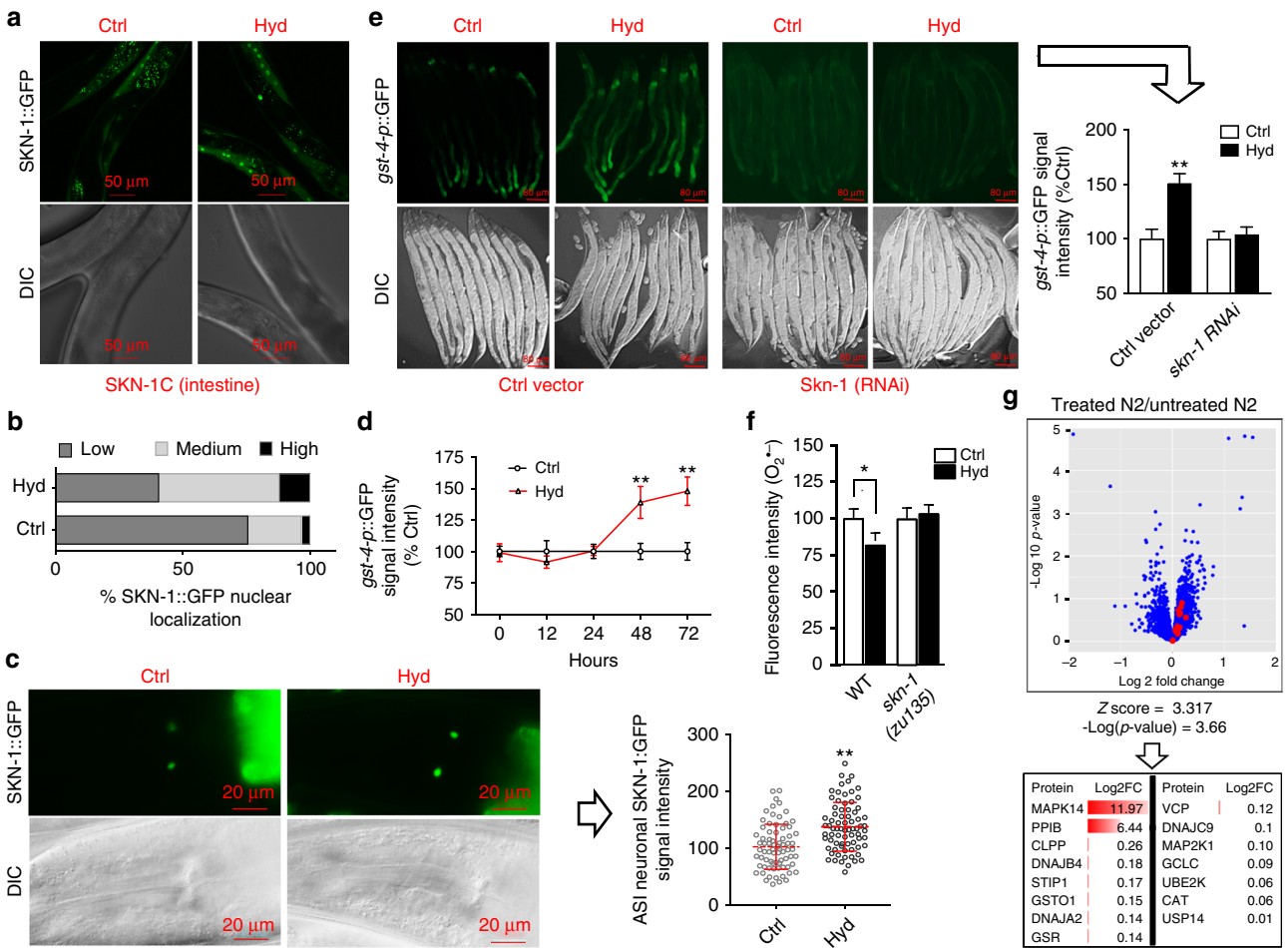

**Fig. 4** SKN-1 pathway is activated with hydralazine treatment in *C. elegans*. In all experiments animals were treated with 100 μM hydralazine for 72 h unless otherwise stated. **a** Fluorescent photomicrograph showing hydralazine treatment increases SKN-1::GFP localization in the intestinal nuclei in *geIs10* transgenic animals. Scale bar = 50 μm. **b** Quantification of SKN-1 intestinal nuclear accumulation represented as percentage of worms with high (≥15 GFP-positive intestinal nuclei), medium (5–15 GFP-positive intestinal nuclei), or low (≤5 GFP-positive intestinal nuclei) nuclear SKN-1::GFP. **p < 0.01, two-tailed Student's *t* test, n = 100 three independent trials. **c** Fluorescent photomicrograph showing GFP signal of SKN-1B in ASI neurons of *ldIs7* transgenic worms. GFP signal intensity in ASI neurons increased with hydralazine treatment as quantified with Image J software. Scale bar = 20 μm. **p < 0.01, two-tailed Student's *t* test, n = 75 four independent trials, mean ± SD. **d** GFP signal intensity quantification showing upregulation of GST-4::GFP in the intestine of hydralazine-treated *dvIs19* transgenic worms after 48 h. **p < 0.01, two-tailed Student's *t* test, n = 35 three independent trials, mean ± SD. **e** Fluorescent photomicrograph showing hydralazine treatment did not increase GST-4::GFP in *skn-1 RNAi* fed *dvIs19* transgenic animals. Scale bar = 80 μm. **p < 0.01, two-tailed Student's *t* test, n = 30 three independent trials, mean ± SD. **f** The level of superoxide ($O_2^{\bullet-}$) measured with DHE fluorophore signal intensity decreased in wild-type *C. elegans* treated with hydralazine but not in SKN-1 mutant worms. *p < 0.05, two-tailed Student's *t* test, n = 30 three independent trials, mean ± SD. **g** A volcano plot showing activation of the SKN-1/NRF2 pathway in wild-type *C. elegans* using Tukey's honestly significant difference test. Proteins were quantified in both treated and untreated animals using label-free mass spectrometry and ratios were uploaded for identification of activated pathways via IPA analysis. The SKN-1/NRF2 pathway was among the activated pathways with *Z* score of 3.317 and *p* value of 0.0002, right-tailed Fisher exact test. A list of human orthologs of SKN-1 pathway members and their Log FC is shown in a table below the plot

fluorescence absorbance corresponding to lipofuscin accumulation and the number of *C. elegans* progeny in populations treated with 100 μM hydralazine. Both parameters showed significant reduction with hydralazine treatment, further supporting hydralazine as a DR mimetic (Fig. 5h, i).

To investigate the role of other pathways involved in aging paradigm, we measured the lifespan in mutant *daf-16 (mu86) C. elegans*, observing ~20% increase in lifespan (Supplementary Fig. 5e). We ruled out induction of ER stress as the hydralazine mechanism of action by tracing *hsp-4p*::GFP protein, a reporter of UPR^ER activation, in worms treated with 100 μM for 72 h by fluorescence microscopic imaging (Supplementary Fig. 5f). We measured HIF1A and HSF1, both known to affect lifespan, in SH-SY5Y cells treated with 10 μM hydralazine. Neither of them

showed a significant change with western blot analysis (Supplementary Fig. 5g).

## Hydralazine NRF2/SKN-1 activation provides neuroprotection. 
To evaluate the potential neuroprotective value of hydralazine we sought to answer two important disease-related questions, (i) will hydralazine-mediated protection against $H_2O_2$ translate to protection against cytotoxicity present in neurodegenerative conditions? (ii) will the extent of hydralazine-induced activation of NRF2 be sufficient to protect cells and *C. elegans* with compromised defense systems? To answer these questions, we used several in vitro and in vivo disease models, (1) mouse primary cortical neuronal cells treated with rotenone, (2) tauopathy model HEK293 cells overexpressing tau residues 244 to

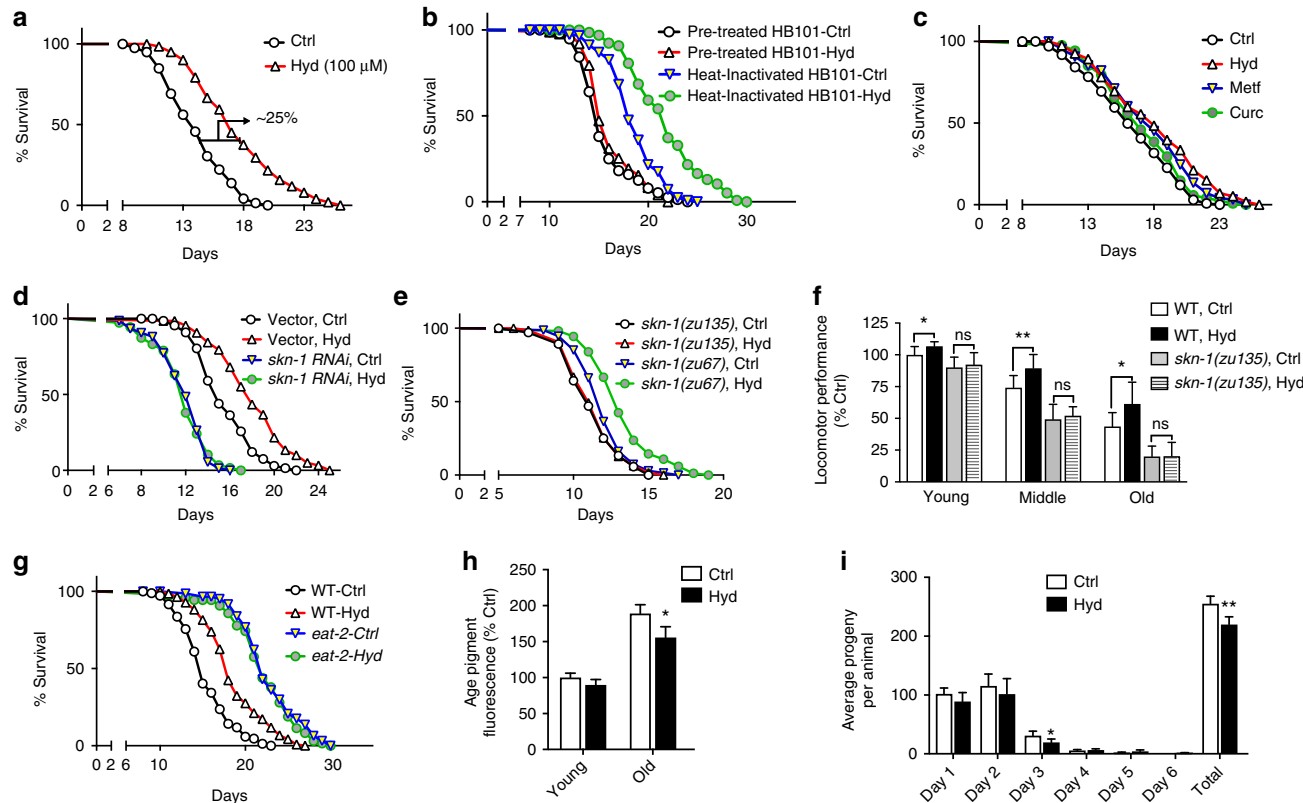

**Fig. 5** Hydralazine extension of healthy lifespan in *C. elegans* is SKN-1 dependent. **a** Hydralazine treatment (100 μM) increased *C. elegans* lifespan by (25%). **b** Pretreatment of bacteria with 100 μM of hydralazine did not extend lifespan in wild-type *C. elegans* but hydralazine treatment (100 μM) significantly extended *C. elegans* lifespan in the presence of heat-inactivated bacteria. **c** Late-stage (at 10 days of age) administration of hydralazine extended the lifespan of wild-type *C. elegans* by ~17%, which is comparable to the other prominent anti-aging drugs such as metformin (20 mM). It is worth mentioning that these comparisons are approximate as it is very difficult to determine how much of each drug entered the worm. **d** Feeding WT worms with *skn-1* RNAi completely eliminated the longevity benefits of hydralazine (100 μM). **e** Hydralazine treatment (100 μM) did not extend lifespan in mutant *skn-1(zu135)*, but expression of functional *skn-1* isoform *b* in *skn-1(zu67)* partially restored the longevity effects of hydralazine. **f** Healthspans of wild-type (WT) and *skn-1 (zu135)* mutant worms were evaluated by measuring locomotor performance in young (5 days for WT, 4 days for mutants), mid-age (10 days for WT, 8 days for mutants) and old (15 days for WT and 12 days for mutants) worms. Hydralazine (100 μm) prevented an age-related decline in locomotion in wild-type worms but not in the *skn-1(zu135)*. *$p < 0.05$ and **$p < 0.01$, two-tailed Student's *t* test, $n = 60$ three independent trials, mean ± SD. **g** Hydralazine treatment (100 μM) did not extend lifespan in *eat-2 C. elegans* (dietary restriction genetic model) indicating partial overlap between dietary restriction and the hydralazine mechanisms of action. **h** Quantification of lipofuscin (age pigment) florescence in wild-type *C. elegans* treated with 100 μM hydralazine showed deceleration of age-dependent accumulation of lipofuscin. *$p < 0.05$, two-tailed Student's *t* test, $n = 90$ three independent trials, mean ± SD. **i** Brood size measurement of wild-type *C. elegans* treated with 100 μM hydralazine showed reduction in progeny production. *$p < 0.05$ and **$p < 0.01$ two-tailed Student's *t* test, $n = 40$ two independent trials, mean ± SD. For all lifespan statistics, see Supplementary Table 2

372, with mutations of P301L and V337M exposed to recombinant tau fibrils that indefinitely propagate tau aggregates (aggregate-positive cells, AP). HEK293 cells growing without forming intracellular tau aggregate were used as control (aggregate-negative cells, AN)[28], (3) wild-type *C. elegans* treated with rotenone, and (4) tauopathy *C. elegans* model expressing anti-aggregate tau (*byIs194*) or pro-aggregate tau (*byIs161*) in the nervous system[29].

We treated primary cortical neuronal cells with 1 μM rotenone as a stressor to study hydralazine-mediated neuroprotection. Rotenone is a pesticide and a mitochondrial complex I inhibitor known to cause PD-like pathology in animal models[30]. The viability of primary neuronal cells stressed with rotenone and treated with 1 μM hydralazine was restored to a level comparable to control cells (Fig. 6a), indicating a complete reversal of rotenone cytotoxicity.

Because SKN-1 plays a key role in *C. elegans* antioxidant machinery, we anticipated a neuroprotective role for hydralazine in *C. elegans* as well[16]. We first assessed the neuroprotective propensity of hydralazine by evaluating its efficacy to protect

against rotenone cytotoxicity. After pretreating adult worms for 72 h with hydralazine (100 μM), worms were treated with rotenone (50 μM) and their viability was measured. Most of the rotenone-treated animals died within 24 h. However, hydralazine-treated animals were significantly protected against rotenone-induced death (Fig. 6b). We also measured locomotion to investigate the health of the rescued animals. The results showed superior locomotor performance in hydralazine pre-treated animals (Fig. 6c). In *skn-1(zu135)* mutants, the locomotor performance of hydralazine-treated worms exposed to rotenone was improved but to a lesser extent (Fig. 6c).

We next used a tauopathy model *C. elegans* strain to study the neuroprotective effect of hydralazine on tau aggregates, a common source of neurotoxicity[29]. The pro-aggregant transgenic strain pan-neuronally expresses highly amyloidogenic and toxic mutated F3ΔK280 fragment of human tau driven by the *rab-3* promoter. The anti-aggregant F3ΔK280-PP transgenic strain was used as the control[29]. Hydralazine treatment (100 μM) significantly extended lifespan of pro- and anti-aggregant strains (Fig. 6d). We also investigated the effect of hydralazine treatment

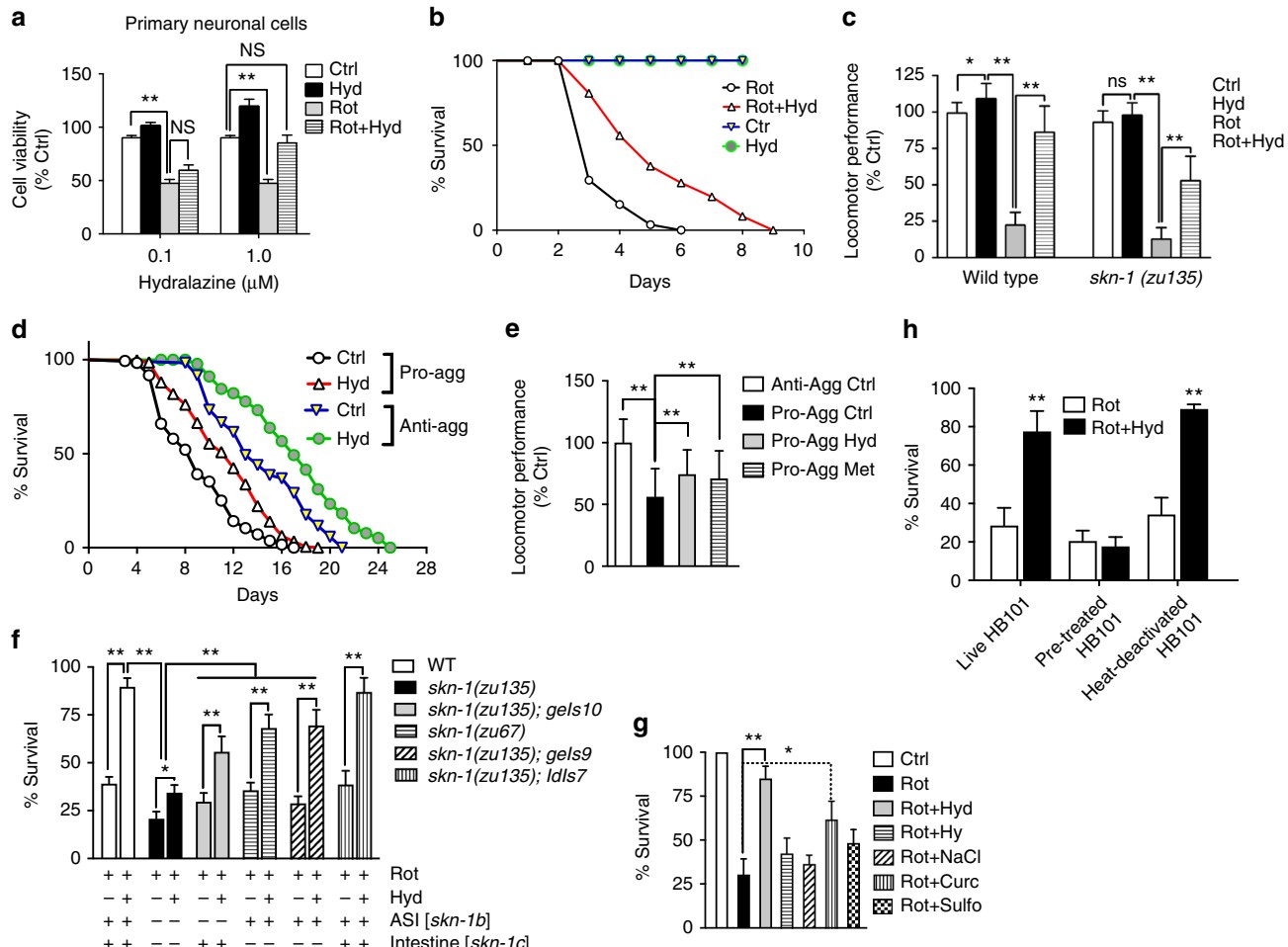

**Fig. 6** Hydralazine protects neuronal cells and *C. elegans* from various stressors. **a** Hydralazine significantly protected primary neuronal cells from rotenone (1 μM) induced death. **p < 0.01, two-tailed Student's *t* test, n = 6, mean ± SD. **b** Hydralazine (100 μM) significantly protected *C. elegans* against rotenone-induced mortality. *p* < 0.0001, Log-rank (Mantel-Cox) test, n = 101. **c** Hydralazine pretreatment (100 μM) prevented rotenone-induced reduction in locomotion of wild-type worms. *skn-1(zu135)* mutants were more vulnerable to rotenone toxicity but still experienced significant protection against rotenone with hydralazine treatment. *p < 0.05 and **p < 0.01, two-tailed Student's *t* test, n = 70 three independent trials, mean ± SD. **d** Hydralazine (100 μM) protected control and tauopathy model *C. elegans* expressing anti-aggregate tau (*byls194*) or pro-aggregate tau (*byls161*) in the nervous system. For statistics see Supplementary Table 2 online. **e** The deficit in locomotor performance of tauopathy model *C. elegans* at day 5 was restored with 100 μM hydralazine treatment. 20 mM metformin was used as positive control. **p < 0.001, two-tailed Student's *t* test, n = 45 three independent trials, mean ± SD. **f** Hydralazine protection (100 μM) was attenuated in zu135 *skn-1* mutant. Presence of a functional copy of SKN-1 isoform C in *geIs10* transgenic animals or SKN-1 isoform B in *skn-1(zu67)* or *geIs9* transgenic animals partially restored hydralazine-mediated protection against rotenone-induced mortality. Integration of SKN-1 isoforms B and C (*ldIs7*) resulted in maximum protection against rotenone-induced death. *p < 0.05 and **p < 0.01, two-tailed Student's *t* test, n = 240 three independent trials, mean ± SD. **g** Worms pretreated with 100 μM of hydralazine or curcumin for 3 days were protected from rotenone-induced toxicity. Worms pretreated with the same concentration of hydrazine, NaCl, or sulforaphane were not. *p < 0.05 and **p < 0.01, two-tailed Student's *t* test, n = 210 two independent trials, mean ± SD. **h** Hydralazine treatment (100 μM) protected *C. elegans* against rotenone cytotoxicity when animals were fed either live or heat-inactivated bacteria but not when they were fed hydralazine pretreated bacteria, indicating a protection mechanism independent of bacterial food source. **p < 0.01, two-tailed Student's *t* test, n = 120 two independent trials, mean ± SD

in restoring the observed deficit in motility of the *C. elegans* tauopathy model. The locomotor performance of the pro-aggregant strain was significantly improved with hydralazine treatment (100 μM). Metformin (20 mM), used as positive control, resulted in a similar effect (Fig. 6e). Activation of the NRF2 pathway, improvement in cell viability, reduction in superoxide concentration, and protection against rotenone were also demonstrated in tauopathy cell model HEK293 aggregate-positive cells treated with hydralazine (Supplementary Fig. 6a–e).

To understand the role of SKN-1 and its different isoforms in hydralazine-mediated neuroprotection, we used two mutants with loss-of-function in *skn-1* and transgenic animals that mosaically express *skn-1* isoform *b* in chemosensory neurons and *skn-1*

isoform *c* in the intestine. In the *skn-1(zu135)* mutant, hydralazine protection sharply decreased compared to wild-type *C. elegans* (Fig. 6f). However, expression of *skn-1* isoforms *b* or *c* partially restored hydralazine-mediated protection. In *skn-1 (zu67)*, better protection was observed compared to the *skn-1 (zu135)*, highlighting the importance of isoform *b*. But activation of both isoforms in transgene *ldIs7* resulted in maximum protection by hydralazine against rotenone-induced cytotoxicity, suggesting a cooperative role for both isoforms (Fig. 6f). To confirm that the protection against rotenone toxicity was the direct result of hydralazine treatment, worms were also treated with equimolar concentration of NaCl or hydrazine which did not result in protection. Treatment with 100 μM curcumin, a

known NRF2 activator, resulted in a much lower protection against the same concentration of rotenone, and no protection against rotenone toxicity was observed in worms treated with 100 μM sulforaphane, another well-known NRF2 activator (Fig. 6g). We also investigated a potential alteration in bacterial metabolism in the presence of hydralazine and its effect on hydralazine-mediated rotenone protection. While hydralazine was protective against rotenone in worms fed with live or heat-inactivated bacteria, animals fed with hydralazine pre-treated HB101 did not experience any protection (Fig. 6h).

We also performed a global comparative proteomics analysis using label-free mass spectrometry to identify pathways activated with hydralazine in worms under rotenone stress. Based on IPA results, the SKN-1/NRF2 pathway was fourth from the top amongst 14 activated stress response pathways in worms treated with hydralazine and rotenone compared to worms treated only with rotenone (Supplementary Fig. 7a). In worms treated with rotenone compared to control worms, SKN-1/NRF2 was not among the activated pathways (Supplementary Fig. 7b). However, when we compared hydralazine-treated rotenone-stressed animals to the control group, SKN-1/NRF2 was found seventh amongst 13 activated stress response pathways, indicating suppression of the SKN-1/NRF2 pathway in rotenone-treated worms (Supplementary Fig. 7c).

## Discussion

At least one new NRF2-activating drug (dimethyl fumarate) has now been approved by the US Food and Drug Administration[31]. Disruption of KEAP1–NRF2 has been suggested as a potential target for Alzheimer's disease[32]. Synthetic triterpenoids activating NRF2-mediated gene transcription have been shown to attenuate neurotoxicity in the 1-methyl-4-phenyl-1,2,3,6-tetra-hydropyridine (MPTP) mouse model of PD[33]. NRF2 has also been identified a therapeutic target for controlling microglial function, activated a result of neural injury or inflammation that participates in progression of PD[34]. NRF2 activation has also been shown to restore lower glutathione in olfactory neurosphere-derived cells from patients with sporadic PD with a malfunctioning NRF2 system[35]. These studies build a case that therapies aimed at NRF2 activation have the potential to provide protection that benefits neurons experiencing age-related degenerative conditions. However, most attempts at finding such a compound have so far failed primarily due to blood–brain barrier issues and lack of mechanistic information[36].

Even though previous findings have connected hydralazine to Alzheimer's disease[37,38], we showed that the protection SH-SY5Y cells experience with hydralazine treatment is not related to the drug's aldehyde chelating properties. Although the chelating properties of hydralazine are highly beneficial. To unravel hydralazine's mechanism of action we used an unbiased proteomic screen to identify cellular pathways modulated by hydralazine treatment (Fig. 2). Multiple pathways were found activated by hydralazine. We pursued the NRF2 pathway due to its direct role in the oxidative stress response and the fact that ROS decreased with hydralazine treatment. Crucial steps in the NRF2 activation process, including NRF2 stability, dissociation of NRF2 from KEAP1, NRF2 phosphorylation and translocation to the nucleus, ARE binding and activation, and upregulation of the NRF2 regulated stress response were monitored in SH-SY5Y cells to confirm the activation of the NRF2 pathway.

We chose C. elegans as a model to address the in vivo relevance of our discovery. During postembryonic stages, the NRF2 ortholog, SKN-1, regulates phase II detoxification genes through constitutive and stress-inducible mechanisms in ASI chemosensory neurons and the intestine, respectively. We showed that

hydralazine upregulates SKN-1, increases its nuclear localization, activates the downstream target GST-4, and reduces the concentration of superoxide in treated worms (Fig. 4).

There are several possible mechanisms for hydralazine-mediated activation of the NRF2/SKN-1 pathway, (1) shifting cellular environment towards a mild pro-oxidant state and altering the conformation of KEAP1 to release NRF2 from a KEAP1–NRF2 complex (hormesis), (2) direct binding and interruption of the KEAP1–NRF2 interaction, and finally (3) activating NRF2 via the PI3K/Akt pathway. Based on the intensity of the oxidative stress measurement probe DCFH-DA, hydralazine does not create a mild pro-oxidant environment and even decreases ROS in a time-dependent manner, which rules out a hormetic response (Fig. 1e). Additionally, we showed that ARE-driven luciferase activity was increased in SH-SY5Y cells treated with hydralazine and antioxidant compound N-acetyl cysteine or Tempol (superoxide dismutase mimetic), indicating a ROS independent activation mechanism (Supplementary Fig. 2c). Results from the KEAP1–NRF2 inhibitor screening assay ruled out the direct binding hypothesis as well (Supplementary Fig. 2d). Considering that NRF2 expression measured by qRT-PCR did not show a significant change with hydralazine treatment but NRF2 signal measured by western blotting and mass spectrometry was increased, it is possible that hydralazine-mediated NRF2 activation is regulated at the posttranslational level (Fig. 3a, h).

NRF2 and its orthologs are known to increase longevity in various organisms from non-vertebrates to vertebrates[21,39]. Increased longevity in diet-restricted C. elegans has been attributed to the NRF2 ortholog SKN-1 activation in the ASI neurons which signal peripheral tissues to increase metabolic activity[18]. On the other hand, skn-1 mutants are sensitive to oxidative stress and have shortened lifespans (25–30%)[17,40–42]. Considering the importance of SKN-1 in C. elegans aging, we chose this model to test the efficacy of early- and late-stage hydralazine treatment on worm lifespan. Our data show that hydralazine extends both median and maximum lifespan of worms by at least 25% with early-stage treatment and ~17% with late stage treatment. SKN-1 was also essential for this action of the drug (Fig. 5a–d). Hydralazine also improved the locomotor performance at all ages (young, middle age, and old) but not in skn-1(zu135), highlighting the importance of SKN-1 activation in improving health in C. elegans (Fig. 5f). Comparing the locomotor performance of treated and untreated wild-type worms, a reduction in age-related locomotor deficiency in hydralazine-treated worms was observed. Our lifespan data with the eat-2 model, taken together with reduction in the accumulation of lipofuscin and reduced fecundity[27,43], suggest that hydralazine is likely a DR mimic. (Fig. 5g–i).

To exclude other longevity modulating pathways, we showed that hydralazine extends lifespan in daf-16 mutant C. elegans, ruling out the involvement of daf-2 insulin/IGF1 signaling (Supplementary Fig. 5e). We also investigated the ER stress response, which is a known NRF2 activator and a conserved prolongevity pathway, by tracing a UPR$^{ER}$ activation reporter protein in worms treated with hydralazine which did not show any activation (Supplementary Fig. 5f)[20]. In addition, we demonstrated that HIF1A and HSF1, other important regulators involved in the aging paradigm, were not upregulated in hydralazine-treated SH-SY5Y cells (Supplementary Fig. 5g).

Hydralazine-mediated extension of the healthy lifespan in wild-type C. elegans raised the possibility that hydralazine might also be effective at protecting neuronal cells from stressors. Activated microglia and astroglia in neurodegenerative diseases release free radicals that harm the neurons. An optimal GSH supply is required to defend against increased free radicals and

maintaining GSH requires activation of the NRF2 pathway[44]. In the hippocampus, one of the brain areas where neurodegeneration starts[45], astrocytes from AD patients have less NRF2 than normal. Decreased glutathione has been reported in the substantia nigra of individuals with PD. Even though NRF2 is localized to the nucleus in the neurons that survive in the substantia nigra, it is not known if the NRF2 transcription machinery is functional[45]. In addition, mutations in the *NRF2* gene have been linked to both AD and PD progression[46,47]. NRF2 is also reduced in motor neurons of the spinal cord and cortex of ALS patients[48]. These studies strongly suggest that the NRF2 system is impaired in individuals suffering from neurodegenerative diseases and health benefits may result from restoration or activation of the NRF2 pathway.

We used multiple in vitro and in vivo models to study the efficacy of hydralazine to counteract neuronal stressors. Hydralazine treatment protected primary neuronal cells from rotenone toxicity, increasing their viability significantly back to levels comparable to the untreated cells (Fig. 6a)[49]. Hydralazine was also effective in improving the cell growth rate of HEK293 cells that form tau fibrils. Because many neurodegenerative diseases are multifactorial disorders[50], we challenged tauopathy model cells with rotenone to mimic conditions closer to what cells may experience under neurodegenerative conditions. We showed hydralazine was capable of protecting cells under stress from multiple stressors (i.e., tau aggregates and rotenone) (Supplementary Fig. 6d). The advantage of this model was that it had a proper control, the same cell line that did not form tau fibrils.

Exposure to rotenone in *C. elegans* is also known to result in significant loss of dopaminergic neurons, a classic feature of Parkinson's disease[51]. Hydralazine also significantly protected wild-type *C. elegans* from rotenone-induced cytotoxicity and improved their locomotor performance in a SKN-1 dependent manner (Fig. 6b, c, g). We measured the lifespan and health of tauopathy model *C. elegans* expressing the F3ΔK280 pro-aggregant tau fragment. Worms expressing the highly amyloidogenic tau species exhibit accelerated tau aggregation and have severely impaired motility and marked neuronal dysfunction[29]. The healthy lifespan of tauopathy model worms was significantly extended with hydralazine treatment which appeared to be aggregate independent (Fig. 6d, e). Hydralazine had the same restorative effect on locomotor performance of tauopathy model *C. elegans* as metformin, a widely accepted anti-aging agent[23].

A common concern in development of new drugs to promote healthy lifespan is side effects as a result of long-term administration. Even though hydralazine is FDA approved and has been in use for decades, change in dosage and duration of treatment may result in previously unknown side effects. However, side effects are less likely to be experienced if the treatment starts at the onset of disease. Only a handful of FDA approved drugs (i.e., rapamycin, metformin, trametinib, and lithium) are known to extend lifespan in model organisms when administrated on mid-late life[21,22,52,53]. Late-stage hydralazine administration, which showed significant efficacy in worms, joins hydralazine with the ranks of FDA approved drugs that induce lifespan extension when commenced at later ages.

While it seems that hydralazine-mediated extension of healthy lifespan in *C. elegans* is primarily SKN-1 dependent, a consideration of our data leaves open the possibility that hydralazine-mediated protection against stress also involves other mechanisms. In *skn-1(zu135)* worms, hydralazine-mediated protection in the presence of rotenone was decreased as a result of SKN-1 inactivation; yet some protection was still observed (~15% compared to wild-type animals ~50%) (Fig. 6f). Similarly, the

locomotor performance of *skn-1(zu135)* worms treated with rotenone and hydralazine was improved but to a lesser extent compared to wild-type animals (Fig. 6c). One possible additional mechanism for the observed protection is change in mitochondrial activation identified by our proteomics screens. A recently published work shows that glycation inhibitors such as hydralazine and metformin change mitochondrial respiration in yeast corroborating this hypothesis[54].

Although hydralazine functions partially like metformin by activating SKN-1 and mimicking calorie restriction[55], there are clear differences between the two in terms of mechanism of action. Metformin-mediated lifespan extension is attributed to altered bacterial metabolism[22] while our data indicate that pro-longevity effects of hydralazine are not related to *C. elegans* food source. It is also reported that metformin acts through a mitohormetic response[56], however our data does not support such a mechanism for hydralazine. It is also worth noting that the anti-aging benefits of hydralazine are observed at lower concentrations compared to metformin, reducing the chance of off-target effects. This is particularly important for long-term administration of the drug for clinical applications.

In summary, we have shown for the first time, using a variety of model systems and assessments, that hydralazine mitigates the impact of cellular stresses on neuronal models by activating NRF2/SKN-1 pathway. We demonstrated that hydralazine performs equally, or better, than other well-established drugs with anti-aging properties. Identifying the direct target(s) of hydralazine binding and unraveling other mechanisms involved in hydralazine-mediated neuroprotection are future directions in our laboratory. We hope our findings will pave the way for the use of hydralazine as a therapeutic compound to treat various aging and neurodegenerative disease models in *C. elegans* and mice.

## Methods

**Chemicals**. All the chemicals were purchased from Sigma (St. Louis, MO) unless otherwise stated.

**Cell culture**. Neuroblastoma SH-SY5Y cells were purchased from ATCC (ATCC CRL-2266) and maintained in DMEM medium supplemented with 10% fetal bovine serum. The cells were cultured in a humidified chamber at 37 °C with 5% $CO_2$. Cells were plated the day before treatment so that the density of cell culture could reach ~70% confluence. Hydralazine was diluted in culture medium from a stock solution. The final concentration of hydralazine and the duration of the treatment were indicated in the text and the Figure legends.

Oxidative stress was induced in cells with different concentrations of stressors (e.g., hydrogen peroxide or rotenone) to test the efficacy of hydralazine. Hydrogen peroxide treatment was done in 5% serum containing medium. At the end of the treatment, cells were collected and washed once in ice-cold PBS buffer, followed by lysis with RIPA buffer (1% Triton X-100, 1% sodium deoxycholate, 0.1% SDS, 0.15 M NaCl, 0.01 M sodium phosphate, pH 7.2) supplemented with cocktails of proteases and phosphatases inhibitors (Thermo Fisher, Waltham, MA) on ice for 1 h with occasional stirring. The cell lysates were then centrifuged at $10,000 \times g$ for 15 min at 4 °C. Supernatants were collected and protein concentration was measured using the bicinchoninic acid (BCA) assay (Pierce, 23228).

**Protein carbonyl assay**. Carbonyl content of hydrazine and hydralazine-treated SH-SY5Y cells with or without $H_2O_2$ stress were measured using Protein Carbonyl Content Assay Kit (Sigma-Aldrich, MAK094). Interfering nucleic acids were removed using 10% streptozocin solution and carbonyl content of the supernatant was measured spectrophotometrically at 375 nm by adding 2,4-dinatrophenylhydrazine (DNPH) followed by detection of dinitrophenyl hydrazine adduct using a micro plate reader following vendor instructions. The protein content of each sample was determined using the BCA assay.

**Cell culture in SILAC media and lysate preparation**. SHY-SY5Y cells were grown in Dulbecco's modified Eagle's medium containing either unlabeled L-Proline, L-arginine (Arg[0]) and L-lysine (Lys[0]) or L-Proline, heavy isotope-labeled L-arginine-$^{13}C_6^{14}N_4$ (Arg[10]) and L-lysine-$^{13}C_6$-$^{15}N_2$ (Lys[8]) (Cambridge Isotope Laboratories, Inc.) supplemented with 10% dialyzed fetal bovine serum (Thermo Fischer, Waltham, MA). Light labeled cells were left untreated to serve as control and heavy

labeled cells were treated with 10 μM of hydralazine for 24 h. After treatment, cells were harvested by trypsinization, washed three times with cold PBS and lysed in a buffer containing 6 M Urea, 2 M Thio-urea, 1% SDS and 100 mM Tris/HCl, pH 8.0 with protease (Thermo Fischer, Waltham, MA) and phosphatase inhibitors. After incubation for 15 min at RT and sonication, the samples were clarified by centrifugation for 15 min at 20,000 × g. Protein content was determined using the 660 nM protein assay kit (Thermo Fisher Scientific, 22660) according to the manufacturer's instructions.

**Protein digestion and peptide fractionation.** Equal amounts of protein from control and hydralazine-treated cells were mixed in a 1:1 ratio and digested in solution. The mix digest (300 μg) was then fractionated into six fractions via strong cation exchange (SCX). SCX cartridges were pre-equilibrated with a buffer composed of 0.5% acetic acid and 2% ACN (wash buffer). The digest was then loaded onto the column and washed with wash buffer and subsequently eluted with a buffer containing ammonium acetate (30, 50, 70, 80, 120, and 500 mM), 0.5% acetic acid, and 2% ACN. Eluted peptide fractions were desalted using reverse phase cartridges.

**Mass spectrometry for SILAC.** All the fractions were analyzed using a Q-Exactive HF mass spectrometer (Thermo Electron, Burlingame, CA) coupled to an Ultimate 3000 RSLCnano HPLC systems (Thermo Electron, Sunnyvale, CA). Peptides were loaded onto a 75 μm × 50 cm, 2 μm Easy-Spray column (Thermo Electron, Sunnyvale, CA) and separated using a 120 min linear gradient from 1 to 28% acetonitrile at 250 nl/min. The Easy-Spray column was heated at 55 °C using the integrated heater. Shotgun analyses was performed using a data-dependent top 20 method, with the full-MS scans acquired at 60 K resolution (at m/z 350) and MS/MS scans acquired at 15 K resolution (at m/z 200). The under-fill ratio was set at 0.1%, with a 3 m/z isolation window and fixed first mass of 100 m/z for the MS/MS acquisitions. Charge exclusion was applied to exclude unassigned and charge 1 species, and dynamic exclusion was used with duration of 15 s.

**Western blot analysis.** Protein expression was determined by western blot analysis. Equal amount of protein from each sample was run in Tris-glycine SDS-PAGE gel, followed by transfer to PVDF membrane. After blocking the membrane with 5% milk for 1 h at room temperature, the membrane was incubated further for 2 h with antibodies specific for target proteins: NRF2 (NBP1-32822, 1/1000 dilution), pNRF2 (S40) (NB100-80012, 1/1000 dilution) from Novus Biologicals (Littleton, CO); KEAP1 (4617, 1/500 dilution), HMOX1 (5061, 1/500 dilution), NQO1 (3187, 1/1000 dilution), HIF1Aα (3716, 1/1000 dilution), HSF1 (4356, 1/1000 dilution), and lamin B1 (12586, 1/1000 dilution) from Cell Signaling (Danvers, MA); GAPDH (sc-47724, 1/5000 dilution) from Santa Cruz (Dallas, TX); β-Actin (MA5-15739-HRP, 1/2000 dilution) from Thermo Fisher (Waltham, MA) and GCLC (ab190685, 1/1000 dilution), GCLM (ab124827, 1/1000 dilution), and TXN (ab26320, 1/1000 dilution) from Abcam (Cambridge, MA). The membrane was subsequently incubated with species-specific HRP-conjugated secondary antibody followed by incubation with chemiluminescence substrate and imaging. The band intensity of each of the target proteins was quantified using either ImageQuant or Image J software (GE Healthcare, Sweden).

**NRF2 knockdown by siRNA transduction.** NRF2 was knocked down using a human NRF2 specific siRNA in a lentiviral vector (sc-37030-V, Santa Cruz, CA). A scrambled siRNA was used as negative control. In total, 50% confluent cells grown in polybrene-containing complete medium were treated with siRNA lentiviral particles directly added into the medium. After 2 days of transduction stable clones were selected using 0.8 μM puromycin for 2 weeks. Expression of NRF2 was determined by western blot analysis as described earlier in this method and by quantitative real-time reverse PCR (qRT-PCR).

**Fluorescence polarization assay.** The direct possible inhibitory effect of hydralazine on a NRF2–KEAP1 interaction was measured using KEAP1–NRF2 Inhibitor Screening Assay Kit (BPS Bioscience, 72020), following the vendor instructions. In total, 25 μl of the prepared master mix, including fluorescently labeled NRF2 peptide, was added to each well in a 96-well plate, followed by adding 5 μl of solution containing corresponding concentrations of hydralazine or sulforaphane as positive inhibitor to each well. The reaction was initiated by adding diluted purified KEAP1 protein. The florescence signal was measured using Spectramax Gemini XPS plate reader (Molecular Devices, Sunnyvale, CA) and data analysis was done as suggested by the vendor.

**Cell viability assay.** Cell growth was analyzed using the MTT cell viability assay. Briefly, at the end of incubation/treatment, MTT reagent was diluted in culture medium and aliquoted into each well. After incubation for 2 h, the medium was aspirated and DMSO was aliquoted into each well to disrupt the cells and dissolve the intracellular MTT dyes. Absorbance was read at 570 nm wavelength in a 96-well plate reader. The absorbance was read at 490 nm in a 96-well plate reader. Primary cortical neuronal cells were cultured in 96 wells plates for three weeks then

treated with 0.1 or 1.0 μM of hydralazine or 1 μM of rotenone for 24 h. Cell viability assay was performed using CellTiter-Glo assay (Promega, G7571).

**Co-immunoprecipitation (Co-IP) and immunoprecipitation.** Cells were lysed in ice-cold 50 mM Tris-HCl buffer with 0.5% Triton X-100 and protease inhibitors, followed by centrifugation at 12,000 × g for 15 min at 4 °C. Protein in the supernatant from each of the samples was incubated with antibody (either specific for NRF2 or KEAP1), at 4 °C overnight, on a rotator with constant stirring. Protein A/G magnetic beads (Thermo Fisher, Waltham, MA) were added into the antibody–antigen mixture followed by incubation for 1 h at 4 °C on a rotator. The tubes were applied to a magnetic stand to collect the beads, followed by washing in lysis buffer for three times. Finally, the bead-bound antibody–antigen mixture was eluted with equal volume of 1× electrophoresis sample buffer. The eluted protein was subjected to western blot analysis as described earlier. NRF2 IP was done using the same antibody used for Co-IP but with a different lysis buffer (150 mm NaCl, 10 mM Tris-HCl (pH 7.4), 1 mm EDTA, 1 mM EGTA (pH 8), 15 Triton X-100, 0.5% NP-40, protease inhibitors). The NRF2-enriched samples were used for WB analysis and mass spectrometry. IP samples were separated on SDS-PAGE and the NRF2 corresponding bands were cut from the gel before in gel digestion and mass spectrometry.

**Cell fractionation.** Nuclear and cytoplasmic fractions were separated using NE-PER Nuclear and Cytoplasmic Extraction Reagents (Thermo Fisher Scientific, 78833). Briefly, cells were harvested and washed with ice-cold PBS buffer. The cells were resuspended in ice-cold CER-I buffer and incubated on ice for 10 min, followed by addition of CER-II buffer and incubation for one more minute on ice. The lysate was centrifuged at 16,000 × g for 5 min. The supernatant was the cytoplasmic fraction. The pellet obtained was lysed in ice-cold NER buffer to obtain the nuclear protein fraction.

**NRF2 transcriptional activity assay.** The transcriptional activity of NRF2 was determined using a luciferase-based transcription activation assay. A vector carrying a NRF2 promoter controlled luciferase gene (firefly luciferase) and a vector carrying the control luciferase (Renilla luciferase) from an ARE reporter kit (BPS Bioscience, 60514) was transiently co-transfected into the cells using Lipofectamine reagents (Thermo Fisher, Waltham, MA). After transduction for 24 h, the cells were treated with hydralazine, hydrazine, N-acetyl cysteine (NAC), Tempol, hydralazine + NAC, or hydralazine + Tempol for another 24 h before being subjected to the luciferase assay with the Dual-Glo Luciferase Assay System (Promega, E2920). Briefly, the cells were incubated with firefly luciferase substrate for 10 min prior to measuring luminescence in a 96-well luminescence plate reader. Subsequently, Renilla luciferase was measured after the addition of Dual-Glo Stop & Glo reagent into the wells with a 10-min incubation. The ratio of luminescence from firefly and Renilla was calculated to normalize and compare NRF2 transcriptional activity.

**Quantitative real-time PCR.** The relative levels of NRF2 and its target genes mRNAs were measured by qRT-PCR. Total RNA was isolated from SHY-SY5 cells using Aurum Total RNA Mini Kit (Bio-Rad, 7326820). cDNA synthesis was done using 1.5 μg of the total RNA following the manufacturer protocol (Maxima First Strand cDNA Synthesis Kit, Thermo Scientific, K1671). RT-PCR reactions were prepared using PowerUp SYBR Green Master mix (Thermo Fisher Scientific, A25742), specific primers (Sigma) of the target genes (Supplementary Table 1) and an equal amount of the diluted cDNAs. Reactions were performed on a C1000 Thermal cycler (Bio-Rad) machine and data were analyzed by Bio-Rad CFX manager 3.1 software using the ΔΔCq method. All data were normalized to the control using actin as internal control.

**Measurement of reactive oxygen species.** Superoxide concentration was measured by dihydroethidium (DHE). Black clear bottom 96 well plates were seeded with about 5000 HEK293 tau aggregate-negative (control) or aggregate-positive model cells per well and about 20,000 SH-SY5Y cells. Cells were treated with hydralazine and the superoxide level was measured by incubating cells with DHE (1 μM) for 30 min. Fluorescence was measured by Spectramax Gemini XPS plate reader (Molecular Devices, Sunnyvale, CA) at 370 nm excitation and 420 nm emission wavelengths. Superoxide concentration in worms was measured after hydralazine treatment for 3 days followed by incubation with DHE (6 μM final concentration) for 30 min. Equal number of animals were transferred to a black bottom 96 well plate and fluorescence was measured as above.

Hydroxyl, peroxyl and other reactive oxygen species (ROS) were measured in SH-SY5Y cells using the Cellular Reactive Oxygen Species Detection Assay Kit (abcam, ab113851) according to the vendor protocol. Tert-butyl hydrogen peroxide (TBHP) was used as positive control to generate ROS.

**Protein profiling using label-free quantitation.** Worms were lysed in 8 M urea, 50 mM Tris-HCl pH 8.0 and 1x protease inhibitor cocktail EDTA-free (Thermo Fisher, Waltham MA) buffer with the aid of sonication. Lysates were centrifuged at 14,000 × g for 15 min at 4 °C and proteins in the supernatant were precipitated

using cold acetone. Disulfide bonds were reduced and alkylated (by DTT and IAA respectively) before diluting the solution to 1.8 M with 25 mM Tris-HCl pH 8.0. Proteins were digested overnight at 37 °C with trypsin (Promega, Fitchburg WI) in the presence of 1 mM CaCl$_2$. Peptides were acidified with TFA and purified using Oasis HLB plates (Waters, UK). A Dionex Ultimate 3000 UHPLC (Thermo Electron, Sunnyvale CA) was coupled to an Orbitrap Fusion Lumos mass spectrometer for the separation and analysis of tryptic peptides. An Easy-Spray column with 75 μM inner diameter and 50 cm long packed with 2 μM C18 material was used for peptide separation. 0.1% formic acid and 2% (v/v) acetonitrile in LCMS grade water was used as buffer A and 10% (v/v) TFA plus 80% (v/v) acetonitrile in LCMS grade water was used as buffer B. In total, 5 μl of sample were injected and separated using a gradient from 0 to 28% mobile phase B over 180 min (240 min total run time). Source voltage was set to 2.2 kV and capillary temperature to 275 °C in the positive ion mode. Ions within the $m/z$ range of 400–1600 were scanned at the resolution of 120,000. Collision induced dissociation method was used to fragment top 10 MS spectra with 2–7 charge states. Label-free quantitation was also performed to detect and quantify NRF2 protein in the treated SH-SY5Y cells. Samples were prepared as described in the IP section. Gel bands were cut in 1 mm cubes and destained using 400 μl of 50 mM TEAB in acetonitrile (1:1) for 30 min at 37 °C. Destained gel pieces were washed with 100% acetonitrile before being reduced and alkylated (by DTT and iodoacetamide respectively). Proteins in the gel pieces were digested overnight at 37 °C with trypsin (Promega, Fitchburg WI). Peptides were extracted from gel pieces using peptide extraction buffer (66.7% ACN; 5% TFA in dH$_2$O) for 15 min at 37 °C and the extract was transferred to a fresh LoBind tube. Gels were washed with 50 μl of 100% acetonitrile and the wash was pooled with the previously collected extracts. Samples were dried down in a vacuum concentrator before clean up using Oasis HLB plates (Waters, UK). MS conditions were the same as above, except the gradient time (120 min instead of 180 min).

**MS data processing and Ingenuity pathway analysis.** LC–MS/MS raw data files were processed using the latest available MaxQuant software (v.1.5.3.30). Proteins were identified by the Andromeda search engine within the MaxQuant program and the search was performed against the UniProt/Swiss-Prot *C. elegans* database. We used one multiplicity as standard label free search. Carbamidomethyl cysteine was set as a fixed modification and methionine Oxidation (M) and Acetyl (protein N-term) were used as variable modifications. The protein and peptide false discovery rates and peptide-to-spectrum match (PSM) false discovery rate (FDR) were set to 1%. Match between runs was performed by using a match time window of 0.7 (minimum) and alignment time window of 20 (minimum). The decoy proteins, known contaminants (after quality control using cluster analysis), proteins identified with a single modified peptide and low confidence proteins identified by only one peptide were filtered out. The p-values for all the statistical analysis were calculated using a two-tailed Student's t test as used for normally distributed data (we pre-processed intensities by binary logarithm) unless otherwise stated. Our entire identified proteins with their fold change values were submitted to IPA and the pathway analysis was performed by the Ingenuity Knowledge Base (genes only) as the reference set with direct and indirect relationships included.

**C. elegans strains and maintenance.** Animals were grown and maintained using standard *C. elegans* conditions at 20 °C on NGM plates and were fed *E. coli* strain HB101. N2 worms were used as wild-type and the following mutants and transgenic strains were used from Caenorhabditis Genetics Center (CGC, University of Minnesota): EU1 *skn-1(zu67) (IV)/nT1[unc-?(n754);let-?] (IV;V)*, EU31 *skn-1 (zu135) (IV)/ nT1[unc-?(n754);let-?] (IV;V)*, CL2166 *dvIs19 [gst-4p::GFP::NLS] (III)*, CL691 *dvIs19 [gst-4p::GFP::NLS] (III); skn-1(zu67)/nT1 [unc-?(n754) let-?] (IV;V)*, LG333 *skn-1(zu135) (IV)/nT1[qIs51] (IV;V); ldIs7 [skn-1b/c::GFP]*, LG348 *skn-1 (zu135) (IV)/nT1[qIs51] (IV;V);geIs9 [gpa-4p::skn-1b::GFP+rol-6(su1006)]*, LG357 *skn-1(zu135) (IV)/nT1[qIs51] (IV;V);geIs10 [ges-1p(long)::skn-1c::GFP+rol-6 (su1006)]*, and LD1 *ldIs7 [skn-1b/c::GFP+rol-6(su1006)]*, DA1116 *eat-2(ad1116)*, BR5270 *byIs161[Prab-3::F3DK280;Pmyo-2::mCherry]*, BR6516 *byIs194;[Prab-3:: F3DK280(I277P)(I308P);Pmyo-2::mCherry]*, CF1038 *daf-16(mu86) (I)*, SJ4005 *zcIs4 [hsp-4::GFP] (V)*.

**Rotenone stress test.** Hydralazine, hydrazine, and NaCl were dissolved in water, while curcumin, sulforaphane and rotenone were dissolved in DMSO. Synchronized L1 larvae were placed on NGM plates preloaded with hydralazine, hydrazine, curcumin, sulforaphane or NaCl. After 3 days young adult worms were transferred to fresh NGM plates either preloaded with rotenone alone or rotenone plus any of the abovementioned compounds. Every worm was subjected to a prodding test with a worm pick every day. A worm was scored as dead when not responding to three repeated proddings. Survival curve was plotted using Prism 7.

**Lifespan analysis.** All lifespan assays were performed at 20 °C using HB101 as food source according to standard protocols[26]. Worms were synchronized by hypochlorite solution. The L1 worms hatched overnight were transferred to agar plates (Corning Inc.). Hydralazine was added freshly from a 5 mM stock to the NGM media. Water was used as control. The L4 population of worms were randomly split to control or treatment groups in a density of about 20–30 worms per

6 cm plate dish. The first day of adulthood was considered day one. In all the experiments with late-onset administration of hydralazine, synchronized L4 animals were moved to the media supplemented with 5′-fluorodeoxyuridine (FUDR, Cayman chemicals) at a final concentration of 40 μM for 2 days to prevent reproduction, then were moved to FUDR-free NGM plates until the final treatments at day 10. The animals that crawled off the plate, ruptured, or died from internal hatching were censored. Worms were transferred to fresh plates every day after reaching adulthood, and every 2 days after reaching 10 days of age. Prodding test as described above was used to count the number of dead worms. Survival curve was plotted using Prism 7 and the significance of the curves calculated by Log-rank (Mantel-Cox) test. To generate inactive bacteria, the HB101 bacteria, grown overnight, were centrifuged, re-suspended in M9, heated 30 min at 65 °C and kept at −20 °C. The bacteria were added (100 μl) freshly to the plates containing hydralazine or water, as control, before transferring animals[57].

**RNA interference.** Synchronized L1 larvae were placed on NGM plates containing 1 mM IPTG and fed HT115 bacterial strain containing scrambled or *skn-1* RNAi plasmids (Julie Ahringer RNAi library clone for *skn-1*: 1568 bp insert, Chr IV 5,652,318–5,653,885, the sequence targeted includes exons 4, 5, and 6 of *skn-1a*, and therefore should target all SKN-1 isoforms). All experiments were done at 20 °C.

**Fluorescence microscopic imaging.** To measure GFP intensity, synchronized populations of worms were anesthetized and arranged on an agarose pad. The intestinal SKN-1::GFP was assayed by confocal microscopy (X40) (Nikon A1R, Nikon Instruments Inc., Melville, NY, USA) (Fig. 4a). The quantification of intestinal SKN-1::GFP was recorded as high (≥15 GFP-positive intestinal nuclei), medium (5–15 GFP-positive intestinal nuclei), or low (≤5 GFP-positive intestinal nuclei) (Fig. 4b). ASI SKN-1::GFP was assayed by a Zeiss AxioImager M2 microscope equipped with a Hamamatsu Flash 4.0 Scientific c-mos camera and Zen2 software (×40) (Fig. 4c). The quantification of ASI SKN-1::GFP was performed with ImageJ using a sliding paraboloid algorithm for reducing the background followed by edge detection. The gst-4p::GFP intensity was measured same as ASI SKN-1::GFP but with ×5 magnification and the quantification was done by ImageJ using the whole worm signal.

**Locomotion assays.** To measure locomotion, worms were subjected to 30 s video recording on a Zeiss Axio Zoom.V16 fluorescence dissecting microscope equipped with Axiocam 503 and ZEN2 software. Bending rate (the number of body-bends-per-second) was measured by placing live animals on a plate containing M9 buffer, filming for 30 s and counting the number of bends. Healthiness of the worms was measured by the bending rate of young, middle age and old animals. For rotenone experiments synchronized N2 or *skn-1(zu135)* L1 worms were placed on NGM plates with 100 μM hydralazine for 3 days. Worms were then transferred to new NGM plates containing either 50 μM rotenone plus 100 μM hydralazine, or 50 μM rotenone for 6 h. The results were obtained from three individual trials.

**Pumping rate assay.** Synchronized day 4 adult wild-type worms were used to count the number of contractions in the pharyngeal terminal bulb as described by Wilkinson et al.[26]. Total number of pumps/min was counted using a hand-held counter under a dissection scope for at least 10 worms on the bacterial lawn and three independent trials were carried out. Worms were treated with hydralazine for 24 h before doing the assay. Mutant *eat-2* animals were used as negative control.

**Bacterial growth rate.** To study the possible growth inhibitory and anti-proliferative effect of hydralazine on HB101 bacteria, they were grown in 96-well plates using liquid LB in the presence of different concentrations of hydralazine. In total, 1–500 dilution cultures of HB101 were grown overnight with shaking at 37 °C. Absorbance (OD 595 nm) was measured every 1 h using a microplate reader.

**Lipofuscin measurement.** Autofluorescent lipofuscin intensity was measured as described[58]. Synchronized populations of young adult and middle age animals were treated with hydralazine for 3 and 10 days, respectively, transferred to 96-well black plates (30 worms per well) and the signal was measured using a Spectramax Gemini XPS plate reader (Molecular Devices, Sunnyvale, CA) at 340 nm excitation and 430 nm emission wavelengths.

**Fecundity assay.** Synchronized L4 animals were transferred to new NGM plates (10 worms per plate) containing H$_2$O as vehicle or hydralazine as treatment. Animals were moved to new plates every day until the end of the reproductive period and the number of progeny in the original plates were counted after 24 h to allow all the fertile eggs to hatch.

**Data availability.** Proteomics data have been deposited in Proteomexchange under accession code PXD005618.

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

## Acknowledgements

We thank Drs. David Mangelsdorf, Melanie Cobb, Peter Douglas, Nima Sharifi, Marc Diamond and Margaret Phillips for advice on our manuscript. We thank Dr. Marc Diamond for providing us with aggregate-positive and negative HEK293 cells and Dr. James Malter for helping us with primary neuronal cell culture and analysis. All *C. elegans* strains were provided by the CGC, which is funded by NIH office of Research and Infrastructure Programs (P40OD010440). This work was supported by the National Institutes of Health (grant R03AG045504 to H.M.), Robert A. Welch Foundation (grant I-1850 to H.M.) and the Cancer Prevention and Research Institute of Texas (grant R1121 to H.M.). H.M. is the founder of Neurodaroo LLC and a member of its scientific advisory board.

## Author contributions

E.D. designed and E.D. and B.S. conducted experiments involving *C. elegans*. Y.Z. and E.D. designed and Y.Z., E.D., B.S., and X.T. conducted experiments involving cell lines. M.G. conducted all IPA analysis. A.H. performed all label-free mass spectrometry. S.Y. conducted the initial carbonyl assays and the SILAC proteomics screen. M.D. conduced the primary neuronal cell studies. S.R., R.L., E.D. and A.C. helped with experimental design and manuscript writing. H.M. designed and supervised the project and wrote the paper.

## Additional information

**Competing interests:** The authors declare no competing financial interests.

