## [Peer Review File · Nature Communications]

Reviewers' Comments:

Reviewer #1:

Remarks to the Author:

NRF2 is a promising drug target for some cancers, and given recent work, has potential as a target for treatments aimed at age related pathologies. This manuscript demonstrates that hydralazine activates the Nrf2/SKN-1 pathway and that this confers resistance to stress as well as a lifespan extension in *C. elegans* at least comparable to that achieved with metformin and curcumin, potentially acting as a DR mimetic. Hydralazine is FDA approved and in use for treatment of hypertension this is a significant finding worthy of publication in Nature Communications. The claims in this paper are novel and will indeed be of interest to others in the field. It has the potential to influence the way people in the field think about SKN-1/Nrfs as a realistic drug target for age-related disorders and is another example of the use of worms as a model for drug testing. In general the claims made by the authors are convincing. Particularly the mammalian cell culture work. I also like to combination of mammalian cells and whole animal models that is demonstrated here. However, I do have concerns regarding some of the *C. elegans* lifespan data (detailed below) that would need to be satisfactorily addressed for the paper to be accepted for publication. The manuscript is clearly written with only some minor errors to correct and some suggestions made as to the order/prioritisation of data (see below). The data is for the most part communicated clearly, although at times there are issues with the interpretation of data being hindered by the lack of appropriate controls (see detailed comments below). The methods is also, for the most part, complete and the statistics satisfactory.

Key points:

- The control strains in all lifespans presented are living unusually short lives. At 20C a median lifespan of 13-14 days is particularly short for an N2/WT strain, which raises the issue of whether the lifespan extension represents a genuine slowing of the onset of age related pathologies or whether the reported stress resistance is protecting against an unidentified issue causing early death.
- A major issue with testing drug efficacy in *C. elegans* is the confounding issue of the effect of the drug on the worm's bacterial food source. The authors do not test the effect of Hydralazine on the bacterial food source. Since the conclusion of this study is that SKN-1 in ASI neurons and thus the DR pathway are required for the life extension induced by this drug, this conclusion is unsafe while the possibility exists that the food source is being modified in some way by the drug. The most worrying scenario being that hydralazine is killing or inhibiting growth of the bacteria and thus indirectly triggering DR pathways. This issue could be resolved by the addition of assays examining the effect of 100uM hydralazine on bacterial growth.
- I noted that the authors used eat-2 as a model to test the relevance of DR here but this may not be the best way to test DR (Zhao et al 2017., Nature Comms) and in addition, skn-1c has also been shown to contribute to eat-2 longevity (Tang and Choe 2015., Mech Aging Dev).
- On a similar topic, the methods note that HB101 is used as the maintenance bacteria for the *C. elegans* strains (this is not standard conditions, those being OP50). I assume that HB101 is also the bacteria used for the lifespan assays (not on RNAi) but this needs to be stated in the methods or legends. Some mention should also be made of what the bacterial food source is for the non RNAi lifespans. Indeed, is the choice of bacterial food the reason for the short lifespans (see first comment)?
- Controls are not always present on individual lifespan plots. This makes statements about transgenes extending or not extending lifespan very hard to interpret (e.g. fig suppl 5C).
- The lifespan table (S3) in the supplement only shows combined data for lifespan repeats. Ideally, the data for each individual trial should be shown and representative experiments presented (with this stated in the legend). This would give a clearer idea of the reproducibility and any variability observed during these treatments.
- The data showing that hydralazine treatment from adult hood extends lifespan is very important, yet it is in the supplement. This data should be mentioned earlier and moved to a main figure. Also, although it is interesting to make the point that its effects are comparable to metformin and curcumin, really it is very hard to compare between drugs as the way these experiments are done

in worms makes it very difficult to know how much drug gets into the animals – perhaps a mention of this in the legend or methods.

- The data presented in Fig 6F is somewhat confusing and raised some questions in my mind: i) *skn-1* RNAi may not function in the neurons (can this be proved?), as RNAi is particularly challenging in this tissue yet hydralazine shows the same effect here as with *skn-1* (*zu135*) in rotenone protection; ii) *skn-1* mutants still have rescued resistance to rotenone in response to Hydralazine but start from a lower level than WT – is the % increase the same? Could a different mechanism be at play here? I think it would be informative to examine the % rescue values here as the baseline for each strain is different.

Minor corrections:

- It would be appropriate to cite Kerr et al., 2017 “Direct Keap1-Nrf2 disruption as a potential therapeutic target for Alzheimer's disease” PLoS Genetics.
- Figure 1- The numbering is incorrect, B and C confused in legend/text.
- There is inconsistency in the quantification of Westerns i.e. some are, some are not.
- A diagram of the *skn-1* isoforms and the alleles used would be helpful.
- “*Daf-16*” suppl fig 5 should be italics and no capitals – also, which allele was used?
- There is no mention of the transgenic strains used in this study being backcrossed to the lab control. Since the transgenic strains are also sourced from different labs this raises the possibility that mutations genetic background of these strains could interfere with the results. However given the consistency of life extension with hydralazine this is perhaps unlikely to be the case.
- Locomotion data interesting, how about rescuing other models of neurodegeneration in worms?

Reviewer #2:

Remarks to the Author:

This study reveals the pro-survival effects of Hydralazine, a drug used for hypertension. The authors show that Hydralazine effectively protects mammalian cells from hydrogen peroxide and rotenone toxicity. In *C. elegans*, they demonstrate that this compound induces longevity and delays age-related motility defects. Confirming their findings in cells, they also show increased resistance against rotenone in *C. elegans*. Using an impressive combination of biochemical assays, proteomics, genetics and imaging, they provide very strong evidence implicating Nrf2/SKN-1 signaling in cells and *C. elegans* as the central mechanism underlying the protective effects of Hydralazine.

Overall this is a well-controlled and thorough study. Interestingly, the effects of Hydralazine seem to be very similar to those of metformin. Like Hydralazine, metformin also prevents carbonylation, advanced glycation end-products, acts through Nrf2 and mimics caloric restriction. In particular, the mechanistic link between SKN-1 and Hydralazine in *C. elegans* is practically identical to a previous study on metformin published by Onken and Driscoll (Metformin Induces a Dietary Restriction-Like State and the Oxidative Stress Response to Extend *C. elegans* Healthspan via AMPK, LKB1, and SKN-1. Brian Onken, Monica Driscoll PLOS One, January 18, 2010 <https://doi.org/10.1371/journal.pone.0008758>). In this published study, Metformin induces a 40% increase in median survival and rescued age-related motility decrease. They showed that metformin acted through SKN-1 (both neurons and intestine) and was DAF-16 independent. Metformin also did not increase the lifespan of *eat-2* mutants. However, it is important to note that a subsequent study by the Gems lab revealed that the main longevity effect related to metformin was mediated through a change in the bacterial metabolism rather than due to SKN-1 activation (<http://dx.doi.org/10.1016/j.cell.2013.02.035>). Therefore, it is essential to test in the current study whether Hydralazine is also acting through the bacteria. This could be done by testing whether Hydralazine pre-treated bacteria modulates *C. elegans* lifespan independently of SKN-1. Ideally, an experiment testing whether Hydralazine has additional beneficial effect compared to metformin would be informative to assess whether they are, at least partly, acting through different mechanisms.

Minor comments:

In the abstract, the authors state that “hydralazine-mediated lifespan extension is SKN-1 dependent with a mechanism most likely mimicking calorie restriction.” This statement could be reinforced by showing reduced lipofuscin accumulation with age and delayed progeny production as seen with metformin.

Figure 2b: Could the authors indicate that the fold changes are represented in log2.

Figure 3d -> e.

Figure 4: 4d: Is this with and without Hydralazine? 4e: Quantification of GST-4p::GFP signal should also be shown for *skn-1(-)* conditions. 4f: Would we not expect a decrease in ROS in the *skn-1* mutants in control conditions?

Line 191: “but did not the same”

Supplementary figure 5c: Considering the importance of this finding and that it is mentioned in the abstract, this panel should be included in a main figure.

Line 259: “both known to effect lifespan” -> affect

Line 593: Figure legend 5: Hydralazine (100mM)-> uM

Concerning the protection by Hydralazin against aggregation toxicity: Figure 6d suggests that the effect is independent of aggregation as the authors note. This is also the case in the tauopathy cell culture model (Supplementary figure 6). Could the authors test whether Hydralazine fails to improve the locomotion of anti-aggregation worms in Figure 6e? This might be a more sensitive test to show an aggregation specific protection. If it is not possible to show an aggregation-specific effect, this should be discussed in the discussion section line 452 to 473.

Discussion:

The differences or similarities between methods of action of Hydralazine versus metformin should be discussed.

Previous findings connecting Hydralazine to Alzheimer’s disease should be mentioned (for example: DOI: 10.1039/C6RA20225J; <https://doi.org/10.1371/journal.pone.0065232>; doi: 10.1021/bi101249p)

Line 422: “deceletrtion” and line 425: “apear”.

Line 430: “ruling out the involvement of IGF signaling”: this should be reduced *daf-2* insulin/IGF-1 signaling.

In the second last paragraph (line 485 to 493), the authors discuss alternative protective mechanisms mediated by hydralazine. The authors should discuss a recent study published by Kazi et al. (<https://doi.org/10.1016/j.jprot.2017.01.015>) showing that glycation inhibitors including Hydralazine extend yeast lifespan and suggest that this effect is caused by changes in mitochondrial respiration. This is particularly interesting in view of the mitochondrial dysfunction pathways that are highlighted by IPA as the third most important change in response to Hydralazine (supplementary table 1). This should be discussed as possible alternative mechanism. Of note, it is unclear why dietary restriction is mentioned in this paragraph (line 493) as this effect is likely to be dependent on SKN-1 b isoform.

The authors should refer in line 489 to figure 6f and in line 491 to figure 6c.

Methods: Line 930: "Rate Assay" -> Pumping

Reviewer #3:

Remarks to the Author:

This manuscript examines the effect of hydralazine on *c. elegans* and cell line models of aging and proteotoxicity. The authors find that hydralazine extends lifespan and provides some protection from the toxicity of proteins associated with aging disease. using an unbiased proteomic approach the authors show that hydralazine acts via the Nrf2/SKN-1 pathway. since hydralazine is already FDA approved, the authors argue that their data recommend it as a potential health-span extension drug for humans.

Overall, the manuscript is written clearly and the protective effect of hydralazine in *c. elegans* models is well demonstrated. Many additional experiments are needed to confirm that this drug should be used on humans, but this is going to be the case for any study. I think that this study would be of interest to the *c. elegans* community.

Response to referees:

Reviewer #1 (Remarks to the Author):

- The control strains in all lifespans presented are living unusually short lives. At 20C a median lifespan of 13-14 days is particularly short for an N2/WT strain, which raises the issue of whether the lifespan extension represents a genuine slowing of the onset of age related pathologies or whether the reported stress resistance is protecting against an unidentified issue causing early death.

To our knowledge lifespan can vary between different labs depending on factors such as temperature, bacterial strain, media composition etc. We agree that our median lifespans are shorter than normal but still they are within the normal range as reported in many other studies published in journals such as *Nature*, *Nature communications* and *PNAS* (PMID: 10819307, 21413243, 17156833, 17978487, 24828042, 28769038, 28627510 and 24889636). The general assumption usually is that aging mechanisms such as Nrf2/SKN-1 pathway activation are universal and can be investigated as long as conditions are standardized (PMID: 19815017). According to 19815017 the mean lifespan of N2 ranges from 12.4 to 22 days in solid nematode growth medium (NGM). Our N2 median lifespans are from 13 to 18 days in different trials which puts them in the normal range for which hydralazine has always shown a pro-longevity effect.

- A major issue with testing drug efficacy in *C. elegans* is the confounding issue of the effect of the drug on the worm's bacterial food source. The authors do not test the effect of Hydralazine on the bacterial food source. Since the conclusion of this study is that SKN-1 in ASI neurons and thus the DR pathway are required for the life extension induced by this drug, this conclusion is unsafe while the possibility exists that the food source is being modified in some way by the drug. The most worrying scenario being that hydralazine is killing or inhibiting growth of the bacteria and thus indirectly triggering DR pathways. This issue could be resolved by the addition of assays examining the effect of 100uM hydralazine on bacterial growth.

We would like to first thank the reviewer by bringing this very important issue to our attention. By doing several experiments now this issue should be addressed. Lifespan experiments and rotenone stress tests were performed in the presence of hydralazine pretreated bacteria and heat-inactivated bacteria shown in Figures 5b and 6h of the revised manuscript. Animals fed with hydralazine pretreated bacteria in hydralazine free NGM media did not experience any protection against rotenone or extension of lifespan. However, hydralazine treatment in the presence of heat-inactivated bacteria still resulted in rotenone protection and extended lifespan. We also studied the effect of 100uM hydralazine on bacterial growth as suggested by reviewer and we did not observe an inhibitory effect (Supplementary Fig. 5b).

- I noted that the authors used *eat-2* as a model to test the relevance of DR here but this may not be the best way to test DR (Zhao et al 2017., *Nature Comms*) and in addition, *skn-1c* has also been shown to contribute to *eat-2* longevity (Tang and Choe 2015., *Mech Aging Dev*).

The underlying mechanisms for longevity induced CR have been heavily investigated but still controversial. There are two major methods to induce CR in *C. elegans*. First by direct restriction in food supply and second by using a genetic model with impaired pharynx pumping (*eat-2* mutant) (19239417). Each method has its own pros and cons. In the first method, to induce CR the concentration of bacteria need to be optimized for each experiment independently. Consequently, the results can vary and hard to reproduce. Also in this method worms usually needs to be grown in liquid culture with varying concentrations of bacteria which is quite cumbersome. However, most aging studies in *C. elegans* have been done under standard culture conditions (i.e., worms are grown on agar plates with lawns of *E. coli* bacteria) (9789046). So, we decided to use the second method. Although *eat-2* is not a perfect model as correctly pointed out by the reviewer, but still offers a universal model that has been widely used in aging studies of *C. elegans* under normal conditions (<https://doi.org/10.1016/B978-0-12-394620-1.00012-6>, PMID: 24828042, 24828042, and 9789046). It is also worth mentioning we only claim that hydralazine is

a partial DR mimetic. Our claim is not based on *eat-2* model alone even though our data showed activation of SKN-1 in ASI neurons which is similar to the model proposed by Bishop and Guarente for CR (17538612). The newly added lipofuscin and fecundity data further support our claim. CR is a complicated issue and more studies need to be done on hydralazine to better understand its CR mimetic effects.

- On a similar topic, the methods note that HB101 is used as the maintenance bacteria for the *C. elegans* strains (this is not standard conditions, those being OP50). I assume that HB101 is also the bacteria used for the lifespan assays (not on RNAi) but this needs to be stated in the methods or legends. Some mention should also be made of what the bacterial food source is for the non RNAi lifespans. Indeed, is the choice of bacterial food the reason for the short lifespans (see first comment)?

HB101 is the bacterial food source in all experiments including lifespan experiments (now added to the method section, as suggested). The major reason for using HB101 were, 1) it is the standard food source for *C. elegans* used in our collaborator's lab (Dr. Rueling Lin). Since we relied on Dr. Lin's lab for all our protocols we preferred to use the same source to avoid protocol reoptimization, 2) although HB101 bacteria is not as widely used as OP50 but to the best of our knowledge there is no report of it being not a good food source for aging research. HB101 is considered a nutrient-rich bacteria based on its superior ability to support wild-type *C. elegans* growth when compared with OP50-fed strain (PMID: 12796460 and 16354781). Shorter lifespan has not been reported for HB101-fed *C. elegans* compared to OP50. It is also reported that lifespan is not markedly affected by bacterial strains OP50, HB101, HT115, and DA837 (PMID: 19844570). The most important point is that all our experiments have been conducted with proper control and the result have always been consistent.

- Controls are not always present on individual lifespan plots. This makes statements about transgenes extending or not extending lifespan very hard to interpret (e.g. fig suppl. 5C).

Controls were added to all plots.

- The lifespan table (S3) in the supplement only shows combined data for lifespan repeats. Ideally, the data for each individual trial should be shown and representative experiments presented (with this stated in the legend). This would give a clearer idea of the reproducibility and any variability observed during these treatments.

Data for individual trials were presented as requested.

- The data showing that hydralazine treatment from adult hood extends lifespan is very important, yet it is in the supplement. This data should be mentioned earlier and moved to a main figure. Also, although it is interesting to make the point that its effects are comparable to metformin and curcumin, really it is very hard to compare between drugs as the way these experiments are done in worms makes it very difficult to know how much drug gets into the animals – perhaps a mention of this in the legend or methods.

The data showing that hydralazine treatment from adult hood extends lifespan has been moved to main figures (**Fig. 5c**). We added text to clarify that these comparisons are approximations for the reasons mentioned above (**line 232-233**).

- The data presented in Fig 6F is somewhat confusing and raised some questions in my mind: i) *skn-1* RNAi may not function in the neurons (can this be proved?), as RNAi is particularly challenging in this tissue yet hydralazine shows the same effect here as with *skn-1*(zu135) in rotenone protection;

We did not observe any lifespan extension in animals fed with *skn-1* RNAi or in *skn-1* mutant (zu135) animals. The animals respond well to *skn-1* RNAi and did not produce viable embryos indicating strong inhibition of *skn-1*. However, to address the reviewer's concern we fed LD1 strain (which expresses GFP-tagged SKN-1) with *skn-1* RNAi and we observed small but significant reduction in ASI GFP signal (figure below). We believe that the mild down-regulation of ASI SKN-1 can mask hyd effect. It is worth mentioning that based on our observation and others (PMID: 11178279, 16317341) neurons are not 100%

resistant to RNAi and their susceptibility varies from neuron to neuron. But because we now know that RNAi treatment probably does not reduce SKN-1B activity to the levels comparable to mutant *zul35* and it was done under a different condition (i.e. different bacterial strain (HT115) and different media composition (presence of IPTG)) compared to the rest of the data in that figure we decided to remove the RNAi data. We appreciate the attention this reviewer has paid to this figure which convinced us to modify the figure to avoid future confusions.

ii) *skn-1* mutants still have rescued resistance to rotenone in response to Hydralazine but start from a lower level than WT – is the % increase the same? Could a different mechanism be at play here? I think it would be informative to examine the % rescue values here as the baseline for each strain is different.

Actually in overall *SKN-1* mutants are more sensitive to any stresses including rotenone so the baseline resistance is lower than wild type. However even in the absence of functional *SKN-1* still we see a little protection (~50% in WT compare to ~15% in mutant). As it discussed in the discussion it seems there are other regulators or pathways for stress resistance against rotenone that are activated by hydralazine. Our proteomics data suggest mitochondria activation is one of the most likely mechanisms involve in the observed protection.

Minor corrections:

- It would be appropriate to cite Kerr et al., 2017 “Direct Keap1-Nrf2 disruption as a potential therapeutic target for Alzheimer's disease” PLoS Genetics.

Kerr et. al were cited as recommended.

- Figure 1- The numbering is incorrect, B and C confused in legend/text.

Figure numbering was corrected.

- There is inconsistency in the quantification of Westerns i.e. some are, some are not.

All our western blots are quantified. The Nrf-2 and reciprocal Nrf-2-Keap-1 western blots were quantified but were not presented as bar plots. Their quantitative values instead were written underneath each lane of the blot with stars signifying their p value.

- A diagram of the *skn-1* isoforms and the alleles used would be helpful.

All *C. elegans* strains carrying mutation or transgenic animals with their alleles are clearly stated in the materials and methods.

• “Daf-16” suppl fig 5 should be italics and no capitals – also, which allele was used?

Corrected and allele was added.

• There is no mention of the transgenic strains used in this study being backcrossed to the lab control. Since the transgenic strains are also sourced from different labs this raises the possibility that mutations genetic background of these strains could interfere with the results. However given the consistency of life extension with hydralazine this is perhaps unlikely to be the case.

As the reviewer alluded to it given the consistency of life extension with hydralazine this is perhaps unlikely to be the case. But we have acquired these strains from CGC and most of strains that we acquired have been backcrossed to N2 animals by the labs who generated the strains.

• Locomotion data interesting, how about rescuing other models of neurodegeneration in worms?

We feel this would be beyond the scope of this manuscript and the data we presented should be sufficient as proof of principal.

Reviewer #2 (Remarks to the Author):

Overall this is a well-controlled and thorough study. Interestingly, the effects of Hydralazine seem to be very similar to those of metformin. Like Hydralazine, metformin also prevents carbonylation, advanced glycation end-products, acts through Nrf2 and mimics caloric restriction. In particular, the mechanistic link between SKN-1 and Hydralazin in *C. elegans* is practically identical to a previous study on metformin published by Onken and Driscoll (Metformin Induces a Dietary Restriction–Like State and the Oxidative Stress Response to Extend *C. elegans* Healthspan via AMPK, LKB1, and SKN-1. Brian Onken, Monica Driscoll PLOS One, January 18, 2010 <https://doi.org/10.1371/journal.pone.0008758>). In this published study, Metformin induces a 40% increase in median survival and rescued age-related motility decrease. They showed that metformin acted through SKN-1 (both neurons and intestine) and was DAF-16 independent. Metformin also did not increase the lifespan of eat-2 mutants. However, it is important to note that a subsequent study by the Gems lab revealed that the main longevity effect related to metformin was mediated through a change in the bacterial metabolism rather than due to SKN-1 activation (<http://dx.doi.org/10.1016/j.cell.2013.02.035>). Therefore, it is essential to test in the current study whether Hydralazine is also acting through the bacteria. This could be done by testing whether Hydralazine pre-treated bacteria modulates *C. elegans* lifespan independently of SKN-

1. Ideally, an experiment testing whether Hydralazine has additional beneficial effect compared to metformin would be informative to assess whether they are, at least partly, acting through different mechanisms.

By doing several experiments this concern has been addressed now. Lifespan experiments and rotenone stress tests were performed in the presence of hydralazine pretreated bacteria and heat-inactivated bacteria. As it represented in the new version of manuscript, animals fed with hydralazine pretreated bacteria in hydralazine free NGM media did not experience any protection against rotenone or extension in lifespan. However, hydralazine treatment in the presence of heat-inactivated bacteria still resulted in rotenone protection and extended lifespan. We also studied the effect of 100uM hydralazine on bacterial growth as suggested by reviewer and we did not observe an inhibitory effect.

Minor comments:

In the abstract, the authors state that “hydralazine-mediated lifespan extension is SKN-1 dependent with a mechanism most likely mimicking calorie restriction.” This statement could be reinforced by showing reduced lipofuscin accumulation with age and delayed progeny production as seen with metformin.

The newly added lipofuscin and fecundity data further support our claim. We thank reviewer #2 for strengthening our manuscript by suggesting to add these data.

Figure 2b: Could the authors indicate that the fold changes are represented in log₂.

Log scale was clarified.

Figure 3d -> e.

Corrected.

Figure 4: 4d: Is this with and without Hydralazine? 4e: Quantification of GST-4p::GFP signal should also be shown for skn-1(-) conditions. 4f: Would we not expect a decrease in ROS in the skn-1 mutants in control conditions?

Quantification was added.

Line 191: “but did not the same”

Corrected.

Supplementary figure 5c: Considering the importance of this finding and that it is mentioned in the abstract, this panel should be included in a main figure.

The data showing that hydralazine treatment from adult hood extends lifespan has been moved to main figures (Fig. 5c).

Line 259: “both known to effect lifespan” -> affect

Corrected.

Line 593: Figure legend 5: Hydralazine (100mM)-> uM

Corrected.

Concerning the protection by Hydralazine against aggregation toxicity: Figure 6d suggests that the effect is independent of aggregation as the authors note. This is also the case in the tauopathy cell culture model (Supplementary figure 6). Could the authors test whether Hydralazine fails to improve the locomotion of anti-aggregation worms in Figure 6e? This might be a more sensitive test to show an aggregation specific protection. If it is not possible to show an aggregation-specific effect, this should be discussed in the discussion section line 452 to 473.

Since hydralazine activates Nrf2/Skn-1 pathway and this is a known and conserved pathway among different organisms, we believe the effects are not aggregate dependent and we expect to see the benefits under different conditions, as represented in the manuscript, not only under a particular aggregate. The aggregate-independence effect of hydralazine was mentioned in the text.

Discussion:

The differences or similarities between methods of action of Hydralazine versus metformin should be discussed.

Detailed discussion was added in the discussion section.

Previous findings connecting Hydralazine to Alzheimer’s disease should be mentioned (for example: DOI: 10.1039/C6RA20225J; <https://doi.org/10.1371/journal.pone.0065232>; doi: 10.1021/bi101249p)

Reference added.

Line 422: “deceletrtion” and line 425: “apear”.

Corrected.

Line 430: “ruling out the involvement of IGF signaling”: this should be reduced daf-2 insulin/IGF-1 signaling.

Corrected.

In the second last paragraph (line 485 to 493), the authors discuss alternative protective mechanisms mediated by hydralazine. The authors should discuss a recent study published by Kazi et al. (<https://doi.org/10.1016/j.jprot.2017.01.015>) showing that glycation inhibitors including Hydralazine extend yeast lifespan and suggest that this effect is caused by changes in mitochondrial respiration. This is particularly interesting in view of the mitochondrial dysfunction pathways that are highlighted by IPA as the third most important change in response to Hydralazine (supplementary table 1). This should be discussed as possible alternative mechanism.

We appreciate reviewer #2 for bringing this point to our attention which we highlighted in our revised manuscript.

Of note, it is unclear why dietary restriction is mentioned in this paragraph (line 493) as this effect is likely to be dependent on SKN-1 b isoform.

The similarities between hydralazine and DR are discussed in details with addition of new data.

The authors should refer in line 489 to figure 6f and in line 491 to figure 6c.

Reference to figures 6f and 6c were added.

Methods: Line 930: “Rate Assay”-> Pumping

Corrected.

Reviewers' Comments:

Reviewer #1:

Remarks to the Author:

The authors have addressed my concerns, particularly those associated with the drug affecting bacterial growth. I also think it was wise to remove the data relating to SKN-1B in the ASI to avoid confusion.

Reviewer #2:

Remarks to the Author:

The authors have addressed all my concerns. The results with the pre-treated bacteria and heat-inactivated bacteria clearly demonstrate that hydralazine is not acting through changes in bacterial metabolism.

Minor corrections still needed: New method section "Fecundity assay" contains a number of typos (Synchnorized, untill, hache); Figure 6g: HyD is now missing in the graph legend next to the grey box.